# Common dysregulation network in the human prefrontal cortex underlies two neurodegenerative diseases

Manikandan Narayanan[1,*], Jimmy L Huynh[2,3], Kai Wang[4], Xia Yang[5], Seungyeul Yoo[3], Joshua McElwee[4], Bin Zhang[3], Chunsheng Zhang[4], John R Lamb[4], Tao Xie[4], Christine Suver[6], Cliona Molony[4], Stacey Melquist[4], Andrew D Johnson[7], Guoping Fan[8], David J Stone[4], Eric E Schadt[3], Patrizia Casaccia[2,3], Valur Emilsson[9,10] & Jun Zhu[3,**]

## Abstract

Using expression profiles from postmortem prefrontal cortex samples of 624 dementia patients and non-demented controls, we investigated global disruptions in the co-regulation of genes in two neurodegenerative diseases, late-onset Alzheimer's disease (AD) and Huntington's disease (HD). We identified networks of differentially co-expressed (DC) gene pairs that either gained or lost correlation in disease cases relative to the control group, with the former dominant for both AD and HD and both patterns replicating in independent human cohorts of AD and aging. When aligning networks of DC patterns and physical interactions, we identified a 242-gene subnetwork enriched for independent AD/HD signatures. This subnetwork revealed a surprising dichotomy of gained/lost correlations among two inter-connected processes, chromatin organization and neural differentiation, and included DNA methyltransferases, *DNMT1* and *DNMT3A*, of which we predicted the former but not latter as a key regulator. To validate the inter-connection of these two processes and our key regulator prediction, we generated two brain-specific knockout (KO) mice and show that *Dnmt1* KO signature significantly overlaps with the subnetwork ($P = 3.1 \times 10^{-12}$), while *Dnmt3a* KO signature does not ($P = 0.017$).

**Keywords** differential co-expression; dysregulatory gene networks; epigenetic regulation of neural differentiation; network alignment; neurodegenerative diseases
**Subject Categories** Genome-Scale & Integrative Biology; Network Biology; Neuroscience

**Mol Syst Biol. (2014) 10: 743**

## Introduction

Different neurodegenerative diseases share similar dysfunctional phenotypes, such as misfolded protein aggregates, neuronal cell death, inflammation, and cognitive decline. Yet, the complexity of these diseases has hindered efforts to obtain a comprehensive view of common molecular mechanisms underlying their initiation or propagation, and thereby hampered development of drugs that could broadly halt neuronal loss in humans (Avila, 2010; Haass, 2010). This study focuses on two such complex diseases in humans, Alzheimer's and Huntington's, for which there is currently no effective intervention to halt or reverse the associated progressive cognitive decline. Late-onset Alzheimer's disease (AD) is the most common form of dementia, accounting for up to 70% of all cases, and is characterized by an initial impact on memory with a subsequent progressive decline in cognitive functioning. The hippocampus and the surrounding cortical regions are the major sites of AD-related pathology, characterized by increasing accumulation of amyloid-beta (Aβ) plaques and tau-related neurofibrillary tangles, both of which are major contributors to the hallmark lesions associated with this disease (Armstrong, 2009). Compared to AD, Huntington's disease (HD) is a rare (~ 5/100,000) neurodegenerative disorder exhibiting cognitive dysfunction and severe motor

1  National Institute of Allergy and Infectious Diseases, Bethesda, MD, USA
2  Department of Neuroscience, Icahn School of Medicine at Mount Sinai, New York, NY, USA
3  Department of Genetics and Genomic Sciences, Icahn School of Medicine at Mount Sinai, New York, NY, USA
4  Merck Research Laboratories, Merck & Co., Inc., Whitehouse Station, NJ, USA
5  Department of Integrative Biology and Physiology, University of California, Los Angeles, CA, USA
6  Sage Bionetworks, Seattle, WA, USA
7  National Heart, Lung and Blood Institute, Bethesda, MD, USA
8  Department of Human Genetics, University of California, Los Angeles, CA, USA
9  Icelandic Heart Association, Kopavogur, Iceland
10 Faculty of Pharmaceutical Sciences, University of Iceland, Reykjavik, Iceland
   *Corresponding author. Tel: +1 301 443 6005; Fax: +1 301 480 1660; E-mail: manikandan.narayanan@nih.gov
   **Corresponding author. Tel: +1 212 659 8942; Fax: +1 646 537 8660; E-mail: jun.zhu@mssm.edu

impairments that arises as a result of dominant mutations within the Huntingtin gene (*HTT*), causing expansion of a polyglutamine region within the HTT protein (Roze *et al*, 2010). However, studies show that other genes and environmental factors can modify the expressivity of the *HTT* polymorphisms in HD (van Dellen & Hannan, 2004). HD pathology features astrogliosis and neurodegeneration of medium spiny neurons, initially affecting the striatum and progressively the cortices and other regions including hippocampus (Roze *et al*, 2010).

Complex diseases and healthy biological systems are increasingly modeled using a network of pairwise interactions among genes, gene products, or biomolecules (Przytycka *et al*, 2010; Barabasi *et al*, 2011), since analyzing the properties of the entire network or subnetworks has the potential to rapidly generate new biological hypotheses, such as uncovering functionally coherent gene modules from co-expression networks (Zhang & Horvath, 2005; Oldham *et al*, 2006) or novel disease genes/pathways (Horvath *et al*, 2006; Chen *et al*, 2008; Emilsson *et al*, 2008; Ferrara *et al*, 2008). We extend this line of research by systematically constructing and analyzing gene dysregulatory networks and underlying molecular interactions that are affected in common between two neurodegenerative diseases, and investigating whether this common network has distinctive features not apparent in the individual disease networks. Co-regulation of genes involved in biological pathways is needed for the proper functioning of a cell, and disruption of these co-regulation patterns has been observed in human diseases such as AD (Rhinn *et al*, 2013; Zhang *et al*, 2013). Detecting such disruptions through a "differential co-expression" (DC) analysis can help us better understand the initiation and propagation of the disease-induced disruptions among interacting genes, compared to commonly used differential expression analysis that simply detects genes whose expression levels change between cases and controls (de la Fuente, 2010; Leonardson *et al*, 2010; Rhinn *et al*, 2013; Zhang *et al*, 2013). We extend this advantage of DC analysis even further by systematically searching for a molecular network of physical (protein–protein and protein–DNA) interactions that connect the identified dysregulation patterns. This is achieved by extending our previous DC analysis (Wang *et al*, 2009) and network alignment (Narayanan & Karp, 2007) methods to identify both the shared dysregulation patterns and the supporting molecular networks affected in two neurodegenerative diseases.

Specifically, we assembled networks of dysregulated gene pairs by analyzing genome-wide gene expression data collected from over 600 postmortem brain dorsolateral prefrontal cortex (DLPFC) tissues of AD and HD patients, as well as non-dementia controls. We focused on the DLPFC brain region as it is commonly affected in both AD and HD (Armstrong, 2009; Roze *et al*, 2010), and our main interest was to understand the common gene regulatory relationships disrupted in degenerative dementia. Gene co-regulation patterns were systematically compared between different groups, and gene pairs whose co-regulation in DLPFC is gained (gain of co-expression, GOC) or lost (loss of co-expression, LOC) in disease cases relative to controls were identified and assembled into the disease-specific differential co-expression (DC) network. Overall, we found GOC gene pairs to be more prevalent than LOC pairs in the DC networks of both neurodegenerative diseases; however, LOC pairs were more consistent across both diseases. Clustering the DC network yielded modules of genes enriched for clinical endpoints

related to brain pathology and dementia, and revealed new disease genes like *FAM59B* (*GAREML*) that participated in LOC pairs at the interface between modules. The AD DC network was replicated in an independent human cohort, before or after exclusion of age-related dysregulation, supporting the validity and robustness of the DC network.

A systematic search for physical (protein–protein and protein–DNA) interactions connecting the DC relations common to AD and HD revealed a 242-gene subnetwork, which was enriched for independent AD, HD and depression related signatures, and revealed an interesting split of LOC/GOC dysregulations among two physically interacting biological processes, neural differentiation and chromatin organization. To test the interconnection of these processes, we constructed two brain-specific knockout mice targeting two genes of similar function in the subnetwork, *DNMT1* and *DNMT3A*. We predicted *DNMT1*, but not *DNMT3A*, as a key regulator for the subnetwork based on the number of their interaction partners, and consistent with our predictions, only the knockout signature of *DNMT1* in the cortex significantly overlapped with the subnetwork genes. This result validates not only the interconnection of two biological processes in the subnetwork but also the difference between our key versus non-key regulator predictions in the subnetwork. In conclusion, our results from inference and analyses of DC networks revealed new insights into the common pathological mechanisms in two neurodegenerative diseases.

# Results

We focused on systematic changes at the molecular level in the dorsolateral prefrontal cortex (DLPFC) from AD patients, HD patients, and non-demented subjects, since this brain region is commonly affected in both AD and HD (Armstrong, 2009; Roze *et al*, 2010). The characteristics of the disease and control samples obtained from HBTRC (Harvard Brain Tissue Resource Center) are summarized in Supplementary Table S1. Briefly, 624 DLPFC (Brodmann area 9) postmortem brain tissues were profiled on a custom-made Agilent 44K array containing 40,638 reporters uniquely targeting 39,909 mRNA transcripts of known and predicted genes. We note that DLPFC tissue samples from 157 HD patients (Supplementary Table S1) represent a significant fraction (~ 1%) of the incidence of HD in the US. All brains were extensively phenotyped for neurohistopathology traits related to AD (Braak stage, specific regional atrophy on a gross and microscopic scale, and ventricular enlargement) or HD (Vonsattel scale severity). The signs of neuropathology were used to confirm diagnoses of AD and HD, as well as the lack of neuropathology in the control group. Finally, all gene expression traits of disease and control samples were adjusted for age, gender, and other covariates (see Materials and methods).

## Identification of dysregulated gene pairs in neurodegenerative diseases

To compare the brain transcriptional networks in AD, HD and non-dementia control brains, we tested for differences in the correlations (co-expression) of all gene pairs computed in each of the groups. Since co-expression of functionally related genes is necessary for the

    

proper regulation of biological processes within a cell and coordination of several cell types that compose a tissue, mapping changes in overall co-expression patterns in disease tissues versus controls could provide indications on which tissue regulatory programs are disrupted by disease. Of particular interest in identifying differentially co-expressed gene pairs is the overall pattern of increasing and decreasing correlations between the brain groups, with gain (or loss) of co-expression, termed GOC (or LOC), indicating increasing (or decreasing) correlation strength in the disease group compared to the control group. For this comparison, we restricted the analysis to DC pairs detected at a 1% false discovery rate (FDR) and that were significantly co-expressed (either negatively or positively) in only one of the two comparison groups (i.e., either cases or controls for GOC or LOC pairs, respectively; see Materials and methods, and Supplementary Dataset D1). We identified 28,223 DC gene pairs (covering 8,897 unique reporters), whose co-expression relationship differed significantly between the AD and non-dementia control groups and was significant in only one of these two groups. Of these identified DC pairs, 65.8% showed gain of co-expression (GOC) in AD, while the rest showed a loss of co-expression (LOC) in AD (Fig 1A and B, Table 1). As cortices are affected in HD as well (Roze et al, 2010), we compared HD samples against non-dementia controls to uncover 106,134 DC gene pairs, almost fourfold more pairs than in AD (Fig 1A and B, Table 1). Figure 1C highlights an example of the pairwise correlation between GPS2 and STARD7 showing similarly disrupted co-expression patterns in both AD and HD.

Similar to what we observed for AD, HD was predominantly characterized by GOC changes compared to LOC changes (Fig 1A and Table 1), suggesting a common pattern of change in the brain network associated with these two neurodegenerative diseases. In fact, 8,776 gene pairs were identified as DC in both AD and HD comparisons to controls, a highly significant overlap comprising 31% of all DC gene pairs and 74% of all DC reporters identified in AD ($P < 2.2 \times 10^{-16}$; Table 1). The overlapping DC pairs always had the same type of disruption (either GOC or LOC) as the control group is same in both comparisons. Furthermore, despite GOC being a more common feature than LOC of the disease networks, the two diseases shared a larger fraction of LOC than GOC gene pairs (Fig 1B and Table 1), which suggests that the LOC changes better reflect the neuropathology common to these diseases.

In both AD and HD comparisons to controls, DC analysis complemented conventional t-statistics-based differential expression (DE) analysis by uncovering additional disease-associated genes. For instance, only 9.9% of the 8,897 DC reporters identified in the AD versus controls comparison (Table 1) overlap with the 2,206 DE reporters, even when the DE is defined using a lenient (uncorrected) P-value cutoff of 0.05 (if we require a DC reporter to participate in at least five DC pairs instead of the one used above, then 12.7% of the 2,667 DC reporters are also DE; see also Supplementary Figs S1, S2, S3, and Supplementary Text A.1). This is in line with the negligible overlap between DE and DC gene sets in peripheral blood from experiments exploring response to feeding in humans (Leonardson et al, 2010), and suggests that mRNA levels of several genes could be buffered against changes in co-regulation patterns and co-regulation patterns could be preserved at different levels of expression of the participating genes.

Several common variants have been shown to associate with AD based on genome-wide association studies (GWAS) catalog (Hindorff et al, 2009), and rare variants in certain genes have also been identified through Mendelian inheritance based on OMIM database (Supplementary Table S2 and Supplementary Text A.2). Among these 23 curated AD-related genes, the allelic effects for APOE, GAB2, SASH1, and FAM113B related genotypes on AD risk were replicated in the HBTRC samples, with the most likely reason for lack of replication of other reported risk variants being lack of power in our relatively small study group (Supplementary Table S2). We find that nine of these 23 AD-related genes show at least one pair of DC in AD (Table 2), and a striking majority (69%) of the DC pairs involving these AD-causing genes showed LOC changes (Table 2), despite the fact that 66% of all DC gene pairs in AD exhibited GOC (Table 1 and Supplementary Text A.2). For example, APOE was involved in 85 DC pairs, all of which were LOC (Table 2). Polymorphism in HTT is the predominant genetic cause of HD (Roze et al, 2010), and we find that HTT gained two co-expression relationships in HD, such as to INSR and NPY1R, which are genes with known links to HD disturbances or progression (Supplementary Text A.2).

As the number of partners of a gene in a transcriptional network can provide clues on the essentiality of the gene and effect on disease (Horvath et al, 2006), we also inspected the hub genes in a network assembled from the DC pairs (Supplementary Fig S4A). The top 10 hub genes (participating in the largest number of DC pairs) in AD had each gained or lost between 144 and 244 co-expression relations in AD compared to controls, whereas these numbers ranged from 339 to 550 for HD versus controls (see Supplementary Text A.3). When considering only disrupted pairs common to both AD and HD, the top 10 hub genes in this shared DC network were RNASE1, GSN, SLC39A11, GPS2, CSRP1, FAM59B, TIMELESS, EZR, AMPD2, and SASH1, six of which were also in the top 10 hub genes for AD (Supplementary Fig S5A–C). Of interest, TIMELESS gained 91 co-expression relations in common in both diseases, and circadian rhythm disruption has been observed in both AD and HD patients (Weldemichael & Grossberg, 2010; Kudo et al, 2011); GPS2, a subunit of the NCOR1–HDAC3 complex involved in anti-inflammation and lipid metabolism (Jakobsson et al, 2009; Venteclef et al, 2010), shared 107 GOC partners between the diseases; and SASH1, which has a known AD association (Heinzen et al, 2009), replicated in the HBTRC samples as already noted (Supplementary Table S2) and shared several LOC partners (more than 80% of its 149 AD, 384 HD, and 136 shared DC pairs were LOC).

## DC patterns are replicated in an independent human dataset

To examine the robustness of the identified dysregulation patterns, we checked whether the DC patterns identified in the AD versus controls comparison showed similar dysregulation in an independent human cohort of late-onset AD and control individuals (Webster et al, 2009). Frontal cortex expression data were available for 31 AD and 40 control individuals in that study. First, the increased number of GOC pairs compared to LOC pairs seen in the HBTRC samples was also observed in the independent dataset at various Q-statistics cutoff values (Fig 2A and B). Next, we checked whether the correlations of the LOC pairs in the control group were robust and could be replicated in the control samples of the independent study. Of the 3,569 LOC pairs that we identified in AD and had both transcripts in a pair represented in the independent dataset, 49.5

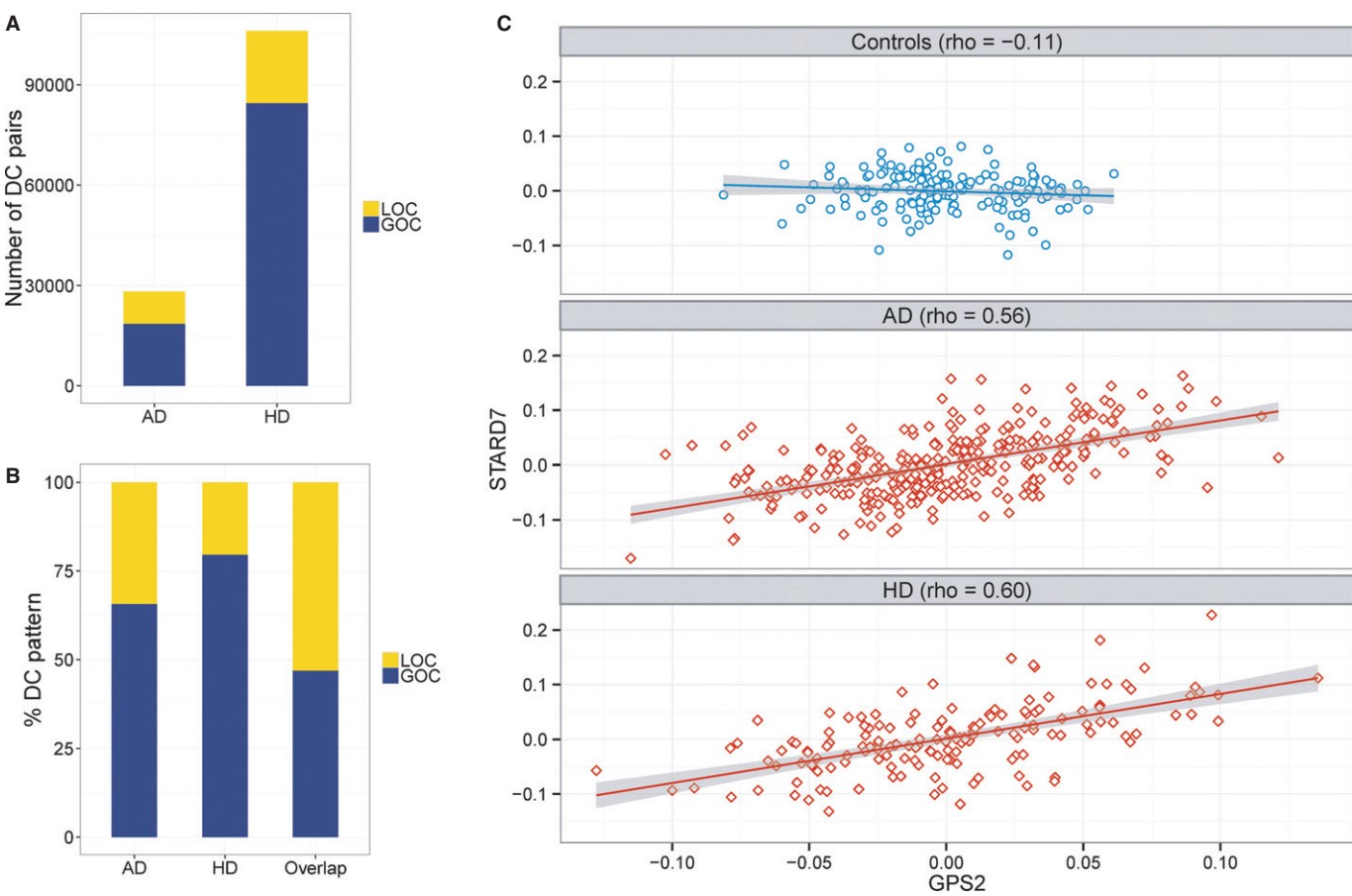

**Figure 1.   Categories of genome-wide, gene–gene dysregulation patterns in neurodegeneration.**
Two categories of changes, gain of co-expression (GOC) and loss of co-expression (LOC), were detected in a genome-wide comparison of gene–gene co-expression relations between neurodegenerative (AD or HD) and normal (non-demented control) brains.

A    There is a greater number (y-axis) of GOC than LOC gene pairs in both AD and HD.
B    Overlapping DC pairs between AD and HD show that LOC is significantly higher in the overlap compared with either disease alone.
C    An example of a gene pair (*GPS2* versus *STARD7*) whose expression variation across individuals (x- and y- axis) reveals a GOC change in both AD and HD.

**Table 1.   Differentially co-expressed (DC) pairs of genes identified via comparison of Alzheimer's disease (AD) or Huntington disease (HD) samples to control samples.**

| Comparison | Q-statistic cutoff (FDR estimate) | Number of DC pairs (number of reporters) | Number of GOC pairs (% of all DC) | Number of LOC pairs (% of all DC) |
|---|---|---|---|---|
| AD versus controls | 25.6 (0.01) | 28,223 (8,897) | 18,560 (65.8%) | 9,663 (34.2%) |
| HD versus controls | 21.7 (0.01) | 106,134 (14,428) | 84,541 (79.7%) | 21,593 (20.3%) |
| Overlap | | 8,776 (6,624) | 4,117 (46.9%) | 4,659 (53.1%) |

and 12.2% were also correlated in the independent control samples using Pearson's correlation $P < 0.05$ before and after Bonferroni correction, respectively (these fractions were 12.5 and 0.4%, respectively, with random pairs of the same size and network connectivity, as obtained by shuffling gene labels; note that proportional test $P = 0$ for both cases).

Finally, we tested whether the magnitude as well as direction of the DC pairs identified in the HBTRC AD set replicated in the independent data. There were 11,561 genes in common between these datasets and an aging dataset discussed below, among which the HBTRC AD set revealed 13,924 DC pairs at $Q > 25.6$ (corresponding to FDR 1% and hereafter called the '*HBTRC-identified*' DC pairs) and the independent AD data yielded 5,913,175 DC pairs at $Q > 3.84$ (analytical $P = 0.05$; we use a lenient $Q$ cutoff for the independent data as it has fewer samples than HBTRC data and is used for replication and not discovery). Of the HBTRC-identified DC pairs, 5.54% got replicated in the independent AD set in the same GOC/LOC direction at $Q > 3.84$ (analytical $P = 0.05$). The much smaller sample size of the independent AD set compared to the HBTRC dataset may explain the low absolute value of this replication rate; however, there is a clear positive trend between signal strength in the HBTRC data and the replication rate

**Table 2. Highlighting well-confirmed genetic causes of AD in the DC network pertaining to AD. We tested replication of published genetic associations to AD in the HBTRC samples and reported the odds ratio (OR), effect allele, and association *P*-values adjusted for age and gender in Supplementary Table S2.**

| Gene | Number of DC gene pairs | %GOC, % LOC | An example of DC gene pair |
|------|-------------------------|-------------|----------------------------|
| APOE | 85 | 0, 100 | APOE–SASH1 |
| PSEN1 | 23 | 0, 100 | PSEN1–GSN |
| PICALM | 1 | 100, 0 | PICALM–CA394907 |
| GAB2 | 3 | 100, 0 | GAB2–MRAP |
| RELN | 5 | 20, 80 | RELN–NCKX3 |
| SASH1 | 149 | 13, 87 | SASH1–CST3 |
| TTLL7 | 9 | 78, 22 | TTLL7–FAM134B |
| BIN1 | 43 | 42, 58 | BIN1–GSN |
| ABCA7 | 70 | 100, 0 | ABCA7–NFKBIA |

(i.e., DC pairs with higher $Q$-values in the HBTRC data are more likely to be replicated in the independent data as shown in Fig 2C), and this replication rate is significantly higher than that of random pairs of the same size as the HBTRC-identified DC pairs (hypergeometric $P = 1.1 \times 10^{-6}$). To further test the impact of network connectivity (inter-relationship) of DC pairs on the replication rate, we randomly selected gene pairs of the same size and network connectivity as the HBTRC-identified DC pairs by shuffling gene labels in the independent data and computed what fraction of them got replicated (Fig 2C, and Supplementary Fig S6C). Repeating this procedure 1,000 times demonstrated that the replication fraction is significant not only for the HBTRC-identified DC pairs (GOC + LOC at $P < 1/1,000$), but also separately for the GOC ($P < 1/1,000$) and LOC ($P < 1/1,000$) pairs. Replication results were similar at other Q cutoffs (2.71 or 6.63 corresponding to analytical $P = 0.1$ or 0.01, respectively) in the independent data (Supplementary Fig S6A). In summary, our set of discoveries as a whole shows significant replication in a cohort of AD and control samples obtained in an external study from different brain banks and profiled using different technologies.

## Most DC patterns are not associated with age

It is worth noting that AD patients in our study are older on average than non-dementia controls (Supplementary Table S1), raising the question of how much age contributes to the dysregulation of HBTRC-identified DC pairs. A neurodegenerative disease state in general, and DC pairs in particular, could result from the normal aging process, accelerated or premature aging induced by AD, or age-independent pathological mechanisms, and disentangling the effect of these factors remains open (Sperling *et al*, 2011) despite some recent advances (Cao *et al*, 2010; Podtelezhnikov *et al*, 2011). To dissect aging effects in our study, we first determined age-associated DC pairs by comparing the expression data of neuropathology-free postmortem samples (Colantuoni *et al*, 2011) of 56 elder (age between 50 and 90 at time of death) to 53 adult (age between 20 and 40 at time of death) group of individuals. Of the HBTRC-identified DC pairs, 32.3% were age-associated—i.e., differentially

co-expressed between the elder versus adult groups even at a lenient cutoff of $Q > 2.71$ (analytical $P = 0.1$). Next, we repeated the replication test using the independent AD dataset as outlined above, but after excluding any age-associated DC pair (20,333,247 DC pairs at $Q > 2.71$ (analytical $P = 0.1$) among genes represented in all three AD/aging datasets) from the HBTRC-identified DC pairs. The results before or after exclusion of age-associated DC pairs were similar both in terms of replication fraction (Fig 2C) and its significance ($P \leq 1/1,000$, 14/1,000, and 1/1,000 for DC, GOC, and LOC pairs, respectively, using the same gene label shuffling test used above; Supplementary Fig S6B and D). These results suggest that most dysregulated pairs we identified in AD were not due to aging but related to the disease itself.

## Modular organization of the DC network elucidates shared pathologies of AD and HD

With confidence that the identified DC pairs are robust, we next aim to understand the biological processes affected by DC pairs in AD and/or HD. Towards this, we attempted to decompose the DC network (Supplementary Dataset D1) defined over thousands of genes into smaller modules of genes, such that genes within each module participated in a larger number of DC relations among themselves than with genes in other modules. By applying a previously published clustering approach (Wang *et al*, 2009) based on spectral techniques and a modularity score function (see Materials and methods), we detected 149 DC modules for AD (Supplementary Fig S4B) and 220 for HD (Supplementary Dataset D2), respectively containing more than 77% of the genes in the DC network for AD and HD.

To understand shared pathologies between AD and HD at the module level, we examined how shared DC pairs were distributed within or between AD modules. We first constructed a network of AD modules by aggregating intra-module DC pairs (both genes in a DC pair within the same module) or inter-module DC pairs (a DC pair interfacing two modules) into weighted links between modules (Fig 3A), and annotated each module as GOC or LOC based on which category was dominant within the module. Among the modules that contained a significant number of shared DC pairs, all but three were LOC modules and they were also grouped together with other LOC modules (Fig 3A) by Cytoscape's 'yFiles Organic' layout algorithm (www.cytoscape.org). This observation is consistent with the shared network being mostly LOC despite the dominance of GOC in the individual disease networks (similar trend was also observed for the HD module network shown in Fig 3B). In enrichment tests done systematically for each module, shared LOC modules M1, M32 in AD, and M24 in HD were significantly enriched for pathways related to metabolism of basic amino acids (Fig 3 and Supplementary Tables S3 and S4), and shared GOC module M6 in AD (along with three other modules) was significantly enriched for genes correlated to an AD clinical endpoint termed Braak stage, which captures the severity of the load of neurofibrillary tangles in the HBTRC samples (Supplementary Table S5).

The overall topology of the DC module network in Fig 3 also revealed widespread loss of co-regulation in the crosstalk (inter-module) relationship between shared DC modules and facilitates hypothesis on regulator genes whose disruption lies at the interface of different modules. For instance, nine genes in the shared LOC AD

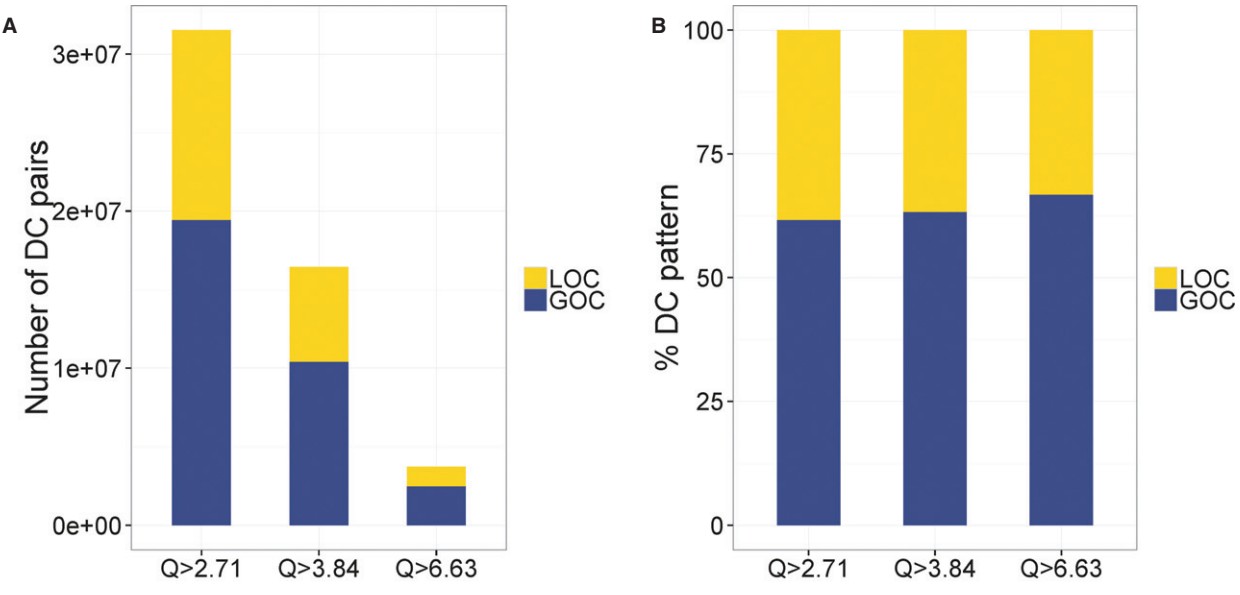

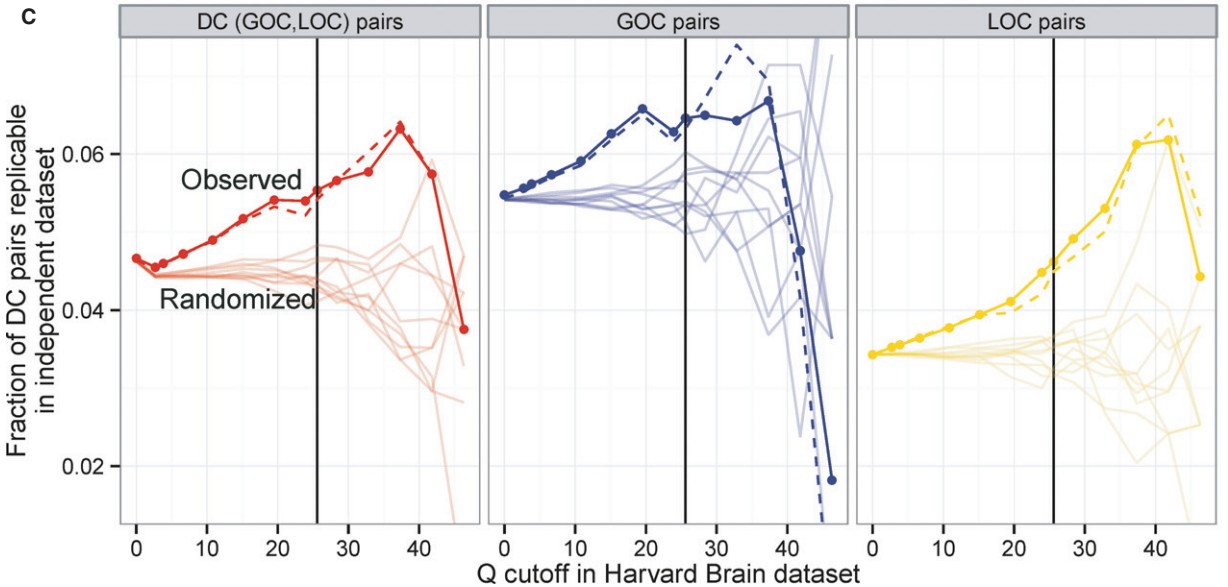

**Figure 2.  Replication in an independent human cohort.**

A, B  The prevalence of GOC over LOC pairs in AD versus controls comparison in an independent human cohort replicates a similar observation in the HBTRC samples. Due to small sample size of the independent cohort, we classified a gene pair as GOC if its' Spearman correlation *P*-value was lower in the AD group compared to the controls and LOC otherwise (thereby relaxing the stringent GOC/LOC definition used in the HBTRC samples).

C  The replication fraction of DC pairs identified in the AD versus controls HBTRC samples (denoted AD DC pairs, and shown as 'Observed' solid lines with dots), and the same replication fraction after excluding any age-associated DC pair from the HBTRC DC pairs (denoted AD-Aging DC pairs, and shown as 'Observed' dashed lines); only DC pairs among genes represented in all three AD/aging datasets were considered. Various cutoffs on *Q* were used in the HBTRC data to derive the DC pairs (with black line indicating the chosen 1% FDR cutoff) and a cutoff of 3.84 (analytical *P* = 0.05) was used in the independent AD data to call when a HBTRC-derived DC pair got replicated in the independent data (note that replication also requires the same GOC/LOC direction in both datasets, with direction in both datasets determined as above using the Spearman correlation *P*-values). The replication fractions of both AD and AD-Aging DC pairs were significant based on 1,000 gene label shufflings (see text and Supplementary Fig S6), random ten of which for the AD DC pairs are shown here as lightly shaded 'Randomized' lines.

module M26 exhibited loss of co-regulation with a single gene *FAM59B* in the GOC AD module M39 in both diseases (Fig 4A). *FAM59B* (also known as *GAREML* or *GRB2 association, regulator of*

*MAPK1-like*) is a gene whose function is poorly characterized; however, its DC relationship with genes in M26 such as *SLC1A2* and *GRIN2C* in the glutamatergic system (whose dysfunction is involved

in neurodegeneration (Sanacora *et al*, 2008)), its hub status and DC partners including *APOE* and *TIMELESS* in the shared DC network (as noted above and in Supplementary Fig S5C), and its correlation with Braak stage in our data (Fig 4B) all taken together support *FAM59B*'s association with neurodegeneration.

**Physical interactions mediating common disruption patterns**

Transcriptional dysregulation in AD and HD could propagate along a network of physical interactions among genes, proteins, and other molecules. To infer such molecular interactions mediating common disruption patterns in AD and HD, we aligned the network of 8,776 DC pairs shared between both diseases (Table 1) with a network of physical interactions compiled from various literature-curated databases such as BioGRID, BIND, MINT, HPRD (Mathivanan *et al*, 2006) totaling 116,220 non-redundant protein–protein, protein–DNA, or other types of pairwise interactions among 12,951 genes. We applied a variant of a rigorous alignment method (Narayanan & Karp, 2007) to find connected regions of the common DC network that were also connected in the physical network, so that disrupted co-expression of these gene pairs was more likely to generate functional consequences (see Materials and methods). The largest aligned subnetwork found by the method comprised 242 genes participating in 401 common DC pairs and 370 supporting physical interactions (Fig 5A and Supplementary Dataset D3; other aligned subnetworks were small with at most two genes). Note that these 242 genes were selected by the algorithm (see Materials and methods) from the background of 1,739 overlapping genes in the physical and common DC network, by virtue of their connectivity in the two networks (i.e., any two of these genes could be connected by a path involving only common AD/HD DC pairs and another involving only physical interactions). The subnetwork was significantly enriched for GO biological processes such as neuron differentiation ($P = 8.8 \times 10^{-7}$) and neurogenesis ($P = 3.1 \times 10^{-6}$), regulation of cellular metabolic process ($P = 1.3 \times 10^{-7}$), gap junction trafficking ($P = 1.1 \times 10^{-6}$), and regulation of apoptosis ($P = 5.6 \times 10^{-6}$). It was also enriched for actin cytoskeleton complex ($P = 6.3 \times 10^{-7}$), and cytoskeletal alterations have been implicated in the disease progression of both AD and HD (Bonilla, 2000; Benitez-King *et al*, 2004).

By providing a scaffold of supporting physical interactions, the 242-gene aligned subnetwork enabled us to hypothesize how disease-induced dysregulation among specific genes could be mediated and propagated. Consider the gene *GSN*, which exhibits the largest number of DC relations in this aligned network and the second largest number of overall common DC relations (117 LOC pairs) shared between both diseases (its immediate neighbors in the aligned subnetwork are shown in Fig 5B). *GSN* encodes the cytoskeletal regulatory protein gelsolin, which is highly enriched in the oligodendrocyte lineage cells (Dugas *et al*, 2006; Cahoy *et al*, 2008; Swiss *et al*, 2011), is increased during the late phase of differentiation of progenitors into premyelinating oligodendrocytes (Swiss *et al*, 2011), and is highly expressed in myelinating cells wrapping the axons (Cahoy *et al*, 2008). Interaction partners of *GSN* include *MAG* (myelin-associated glycoprotein), a molecule synthesized in myelinated oligodendrocytes and localized at the axonal interface (Trapp *et al*, 1989; Arroyo & Scherer, 2000); *GJB1* encoding the protein Connexin 32 that localizes in the myelinated fibers of the

central nervous system (Scherer *et al*, 1995); and *SOX10*, a key transcriptional regulator of myelination in both central and peripheral nervous systems (Stolt *et al*, 2002). The relation of these interaction partners of *GSN* and *GSN* to oligodendrocytes suggests that alterations in the transcriptional programs or composition of oligodendrocytes in the prefrontal cortex could be a common feature of AD and HD.

The aligned subnetwork, consisting of gene pairs commonly dysregulated in AD and HD in the HBTRC data, overlaps with independent AD, HD, and other brain diseases signatures. For instance, the subnetwork contains 3 AD GWAS or OMIM genes (*APOE*, *PSEN1*, and *BIN1*, $P = 0.0029$) and significantly overlaps with genes known to be upregulated in brain samples from AD patients (Blalock *et al*, 2004) ($P = 9.5 \times 10^{-12}$). When comparing siRNA candidates that suppress *HTT* toxicity in a HD model (Miller *et al*, 2012), seven of the 147 siRNA candidates represented on our expression microarray (*ASGR1*, *CAPN2*, *DAXX*, *FABP5*, *RAP1A*, *RNF130*, and *TRPV6*) mapped to the subnetwork ($P = 0.0027$). Finally, the 242-gene aligned network overlapped with genes downregulated in postmortem brains of major depressive disorder patients ($P = 5.1 \times 10^{-15}$), which also comprises myelination and signal transduction-related genes (Aston *et al*, 2005). These results together suggest that the aligned subnetwork could underlie transcriptional disruptions that occur in multiple neural diseases.

**Dichotomous dysregulation of two biological processes in the aligned subnetwork**

To glean insights into how the 242-gene aligned subnetwork found above could underlie multiple brain diseases, we inspected the topological distribution of the GOC, LOC, and physical interactions across the network. We observed a surprising LOC/GOC dichotomy that shed light on the epigenetic dysregulation of oligodendrocyte differentiation and myelination in AD and HD. We specifically found that the 151 LOC genes and the 103 GOC genes in this subnetwork were largely distinct with an overlap of only 12 genes (a LOC gene is loosely defined as any gene with at least one LOC/green edge and a GOC gene as any gene with at least one GOC/red edge in Fig 5A), and these two largely distinct gene groups were connected by numerous physical interactions (165 of total 370 shown as black edges in Fig 5A). The inferred dysregulated processes in these groups revealed an interesting picture: within the whole 242-gene subnetwork, 33 genes were linked to neuronal differentiation ($P = 8.8 \times 10^{-7}$ as seen above), and almost all of them (29) were LOC genes. When considering only the LOC genes in the subnetwork, there was even more significant enrichment for neuronal differentiation genes ($P = 1.8 \times 10^{-9}$), and for genes regulated during oligodendrocyte differentiation (Swiss *et al*, 2011) ($P = 1.02 \times 10^{-20}$). The LOC genes overlapped specifically with gene clusters that were upregulated in a non-transient fashion during late stages of oligodendrocyte differentiation ($P = 8.5 \times 10^{-4}$, $1.6 \times 10^{-4}$, $0.0014$, $3.04 \times 10^{-10}$, $2.4 \times 10^{-4}$ and $1.2 \times 10^{-11}$, respectively, for clusters 7–12 in (Swiss *et al*, 2011)). On the other hand, the GOC genes in the subnetwork, including *GPS2*, *DNMT1*, *DNMT3A*, *YY1*, *HDAC5*, *HIST2H3A*, and more, were enriched for GO biological processes negative regulation of gene expression ($P = 1.2 \times 10^{-7}$) and chromatin organization ($P = 6.6 \times 10^{-7}$) (Fig 5A).

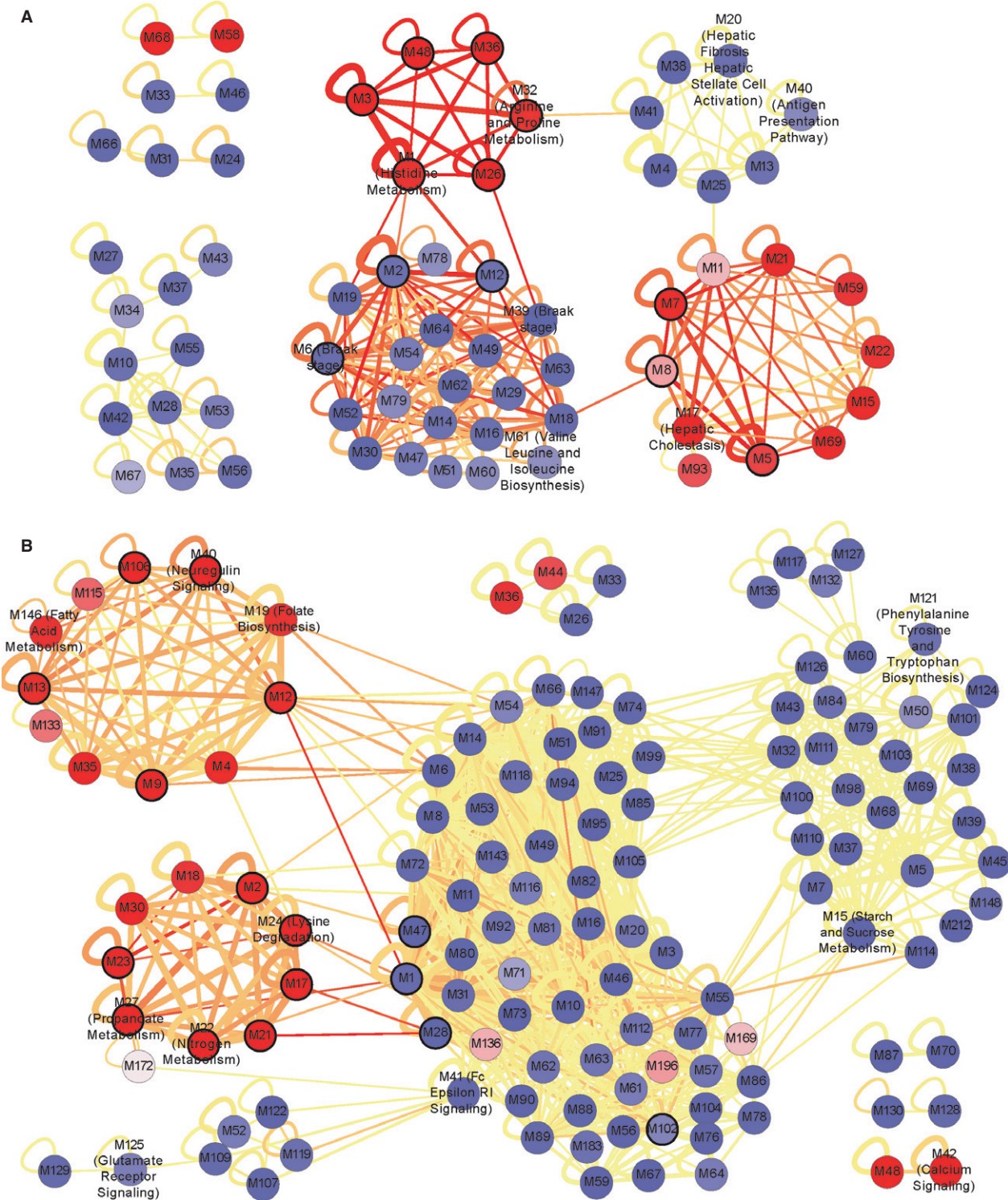

**Figure 3.   Overall topology of shared dysregulation in AD and HD.**

A, B   Topology of the DC network among the AD modules (A) and HD modules (B) reveals a significant enrichment of shared DC pairs in more LOC than GOC modules, and functional/clinical annotations of several modules. A self-loop edge indicates intra-module DC pairs. The thickness and redness of an edge scales with the number of aggregated DC pairs and the fraction of these pairs shared between the two diseases, respectively. A module with dark border is significantly overrepresented for shared intra-module DC pairs (hypergeometric $P < 0.05$ after Benjamini–Hochberg adjustment for multiple testing), and a module's color indicates whether it comprises predominantly GOC (blue) or LOC (red) pairs. Only modules with connections to other modules and edges aggregating at least 20 DC pairs are shown. Any module enriched for a pathway at hypergeometric $P < 0.05$ (after Bonferroni correction for the pathways tested) is labeled by the most enriched pathway, and modules enriched for genes correlated to AD Braak stage severity are also labeled.

   

Disordered chromatin organization and related epigenetic mechanisms of histone modifications and DNA methylation are increasingly appreciated as key pathogenic factors for AD and HD, but there is still much research to be done for instance in terms of human studies of DNA methylation changes in AD, as they are scarce and based only on small cohorts of individuals (see reviews (Balazs *et al*, 2011; Coppedè, 2013; Jakovcevski & Akbarian, 2012)). Our study based on hundreds of human post-mortem brains provided a unique view, as noted above, of transcriptional dysregulation of chromatin modifier genes (including methylation-related genes like *DNMT1* and *DNMT3A*) in neurodegeneration and their interconnections in the aligned subnetwork to oligodendrocyte differentiation genes such as *SOX10* and *GSN*. Hyper-methylation of the key oligodendrocyte-specific transcription factor (TF) *SOX10* has been linked to oligodendrocyte dysfunction (Iwamoto *et al*, 2005), and we have shown before that histone modifications of *GSN*—with a large number of LOC connections in the subnetwork as noted above—contribute to oligodendrocyte differentiation *in vitro* (Liu *et al*, 2003). We have also shown that age-dependent histone deacetylation controls

oligodendrocyte differentiation (Shen *et al*, 2008). All these results suggest that the 242-gene subnetwork involving two interacting biological processes, loss of co-regulation in oligodendrocyte differentiation or myelination and gain of co-regulation in chromatin organization, could underlie multiple neurodegenerative diseases.

### Validating epigenetic regulation of neural differentiation and *DNMT1* as a key regulator in the aligned subnetwork

Among genes involved in chromatin organization in the 242-gene aligned subnetwork, *GPS2* and *DNMT1* are top hub genes with 18 and 16 connections within the subnetwork, respectively. *GPS2* is a subunit of the NCOR1–HDAC3 complex, and we have shown that histone deacetylation controls oligodendrocyte differentiation (Marin-Husstege *et al*, 2002; Shen *et al*, 2008). DNA methylation by DNMT1 or DNMT3A enzymes has been broadly implicated in neural development and differentiation (Takizawa *et al*, 2001; Wu *et al*, 2010, 2012) as well, but here we aim to validate whether DNA methylation regulates oligodendrocyte differentiation genes in the

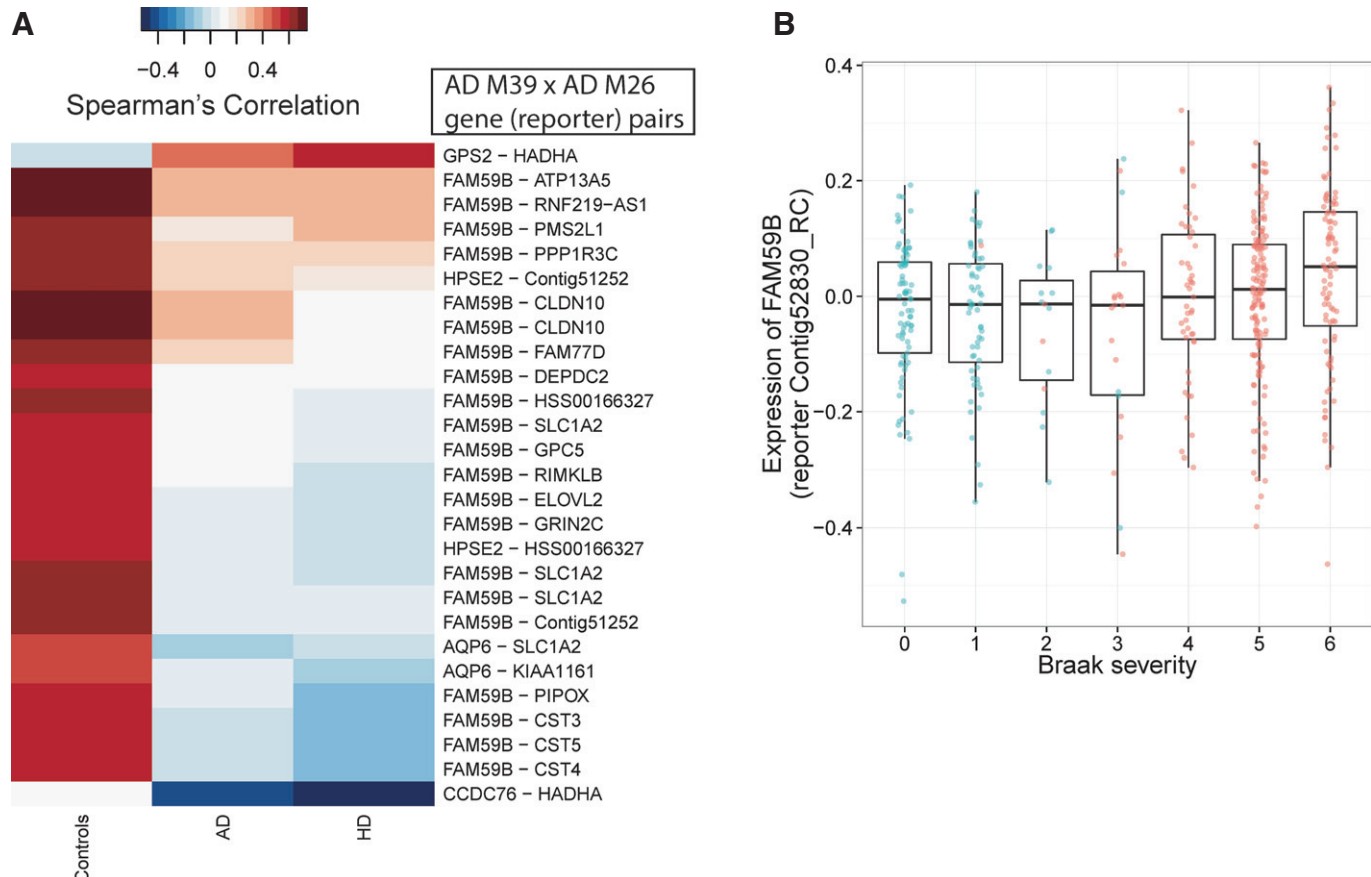

**Figure 4.   Shared crosstalk between two DC modules reveals a new neurodegenerative association.**

A   The crosstalk (inter-module) DC relations between AD GOC module M39 and AD LOC module M26 are dominated by the loss of co-regulation of a single gene *FAM59B* in M39 with several genes in M26. Note that genes represented by multiple reporters appear more than once in the heatmap.

B   Expression of a *FAM59B* reporter correlates with Braak severity score (*P* = 0.00095) across all AD and control DLPFC samples (shown as jittered red and blue dots, respectively).

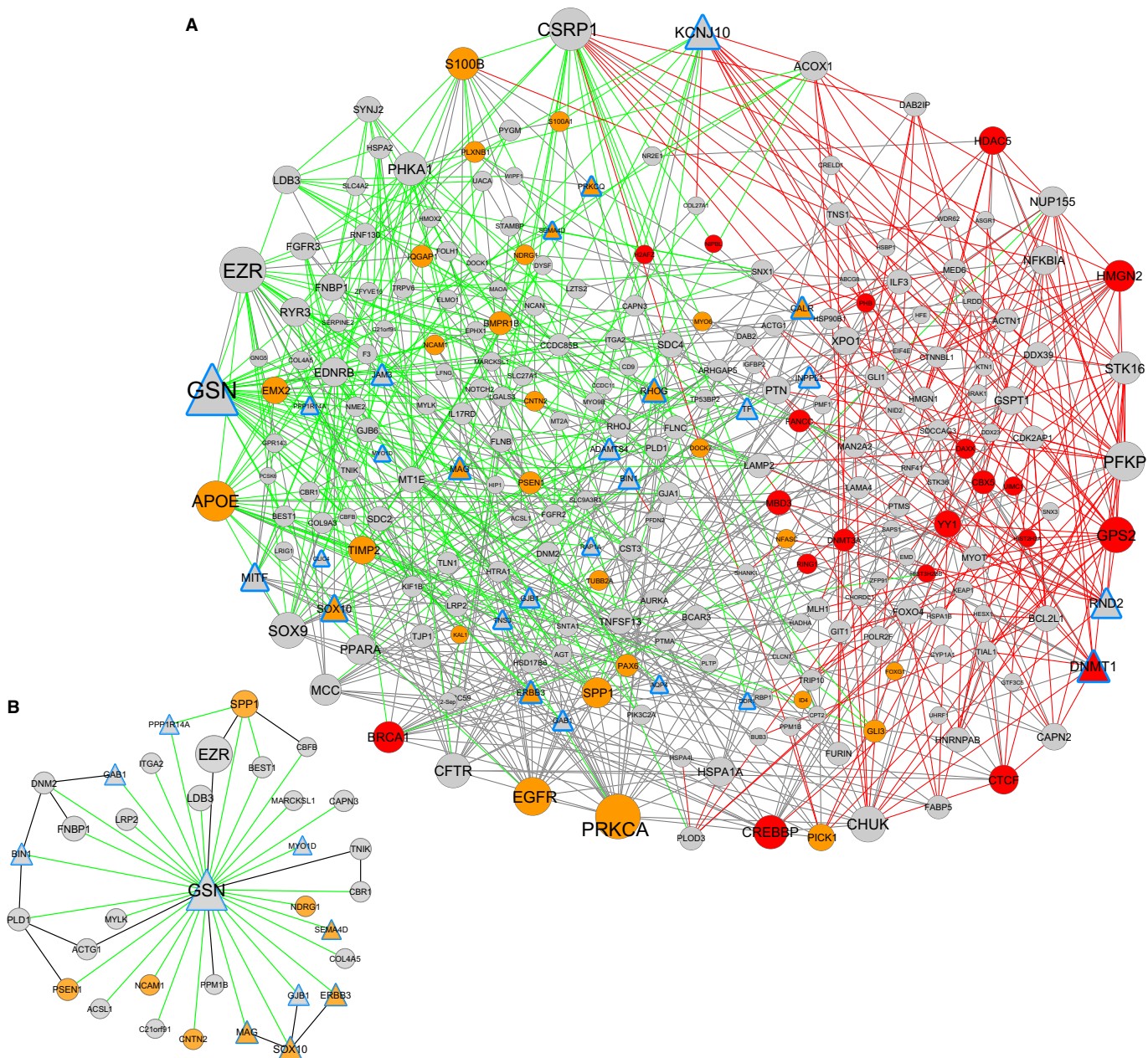

**Figure 5. Physical network links regions of GOC disruption in chromatin organization and LOC disruption in neuronal/oligodendrocyte differentiation.**

A, B We systematically aligned the network of common DC pairs detected in both AD and HD with the network of literature-curated physical (protein–protein and protein–DNA) interactions to obtain a subset of genes that is maximally connected in both networks. This aligned subnetwork is shown in (A), and the immediate neighborhood of gene *GSN* in this aligned subnetwork is shown in (B). Red and green edges are, respectively, the GOC and LOC pairs among the common DC pairs, and black edges mark the physical interactions. Genes with LOC (green) edges were significantly enriched for genes involved in neuronal differentiation (nodes in orange), and genes with GOC (red) edges were significantly enriched for genes involved in chromatin organization (nodes in red). The node size is proportional to the number of node's links in the subnetwork. *Dnmt1* brain-specific knockout signature significantly overlaps with the 242-gene subnetwork (A) ($P = 3.1 \times 10^{-12}$) and the immediate neighborhood of gene *GSN* in this aligned subnetwork (B) ($P = 8.4 \times 10^{-10}$). Triangle-shaped nodes with blue borders are genes differentially expressed in the *Dnmt1* brain-specific knockout compared with wild-type littermates.

242-gene subnetwork disrupted in AD and HD. There are two DNA methyltransferases, *DNMT1* and *DNMT3A*, in this disease subnetwork with 16 and 6 interactions, respectively. *DNMT1* being one of the top hub genes with many interaction partners is likely to play a key regulatory role in the subnetwork, whereas *DNMT3A* with few

interaction partners is likely to play a smaller role. To test these predictions, we generated two oligodendrocyte-specific conditional knockout (CKO) mice, *Dnmt1* CKO and *Dnmt3a* CKO, and dissected cortices from these brain-specific knockout and respective littermate control mice for profiling using RNA-seq technology (see Materials

and methods). Analysis of this data yielded 388 genes that were significantly differentially expressed in *Dnmt1* CKO mice compared to their littermate controls at 10% FDR (the *Dnmt1* CKO signature), and 42 genes in the *Dnmt3a* CKO signature (see Materials and methods, and Supplementary Datasets D4 and D5). Consistent with our predictions, the *Dnmt1* CKO signature included key oligodendrocyte differentiation or myelination genes (including the top hub gene *GSN*, the TF *SOX10*, *MAG*, *GJB1*, and others discussed above), and significantly overlapped with the entire disease subnetwork as well as the *GSN* local subnetwork ($P = 3.1 \times 10^{-12}$ and $8.4 \times 10^{-10}$, respectively, as shown in Fig 5). Broadly, the *Dnmt1* CKO signature was enriched for genes involved in GO biological processes, nerve ensheathment, glial cell differentiation, nerve maturation, and lipid biosynthesis ($P$-values $1.37 \times 10^{-11}$, $3.4 \times 10^{-10}$, $3.2 \times 10^{-9}$, and $7.2 \times 10^{-9}$, respectively), again consistent with the biological functions of the subnetwork. Moreover, the *Dnmt1* CKO mice showed increased predilection to seizures, an incidence also increased in patients with AD (Amatniek *et al*, 2006) as well as juvenile form of HD (Cloud *et al*, 2012). In contrast, the *Dnmt3a* CKO signature was much smaller, only marginally overlapped with the subnetwork ($P = 0.017$), and was not enriched for any GO biological process. These results not only validate the interconnection between the two dysregulated biological processes in the disease subnetwork, but also validate our key regulator predictions.

# Discussion

We show for the first time that the global pattern of gene–gene co-regulation in the human brain cortex is drastically altered in a shared fashion in neurodegenerative diseases like AD and HD, by employing a systematic differential co-expression (DC) analysis that complements conventional differential expression analysis for finding disease-associated changes. The disrupted DC patterns we found either can echo reactive responses to the neuronal pathology associated with neurodegenerative disease or may indicate a direct causal relationship with the disease. We found that GOC between pairs of genes was a more dominant feature than LOC in AD and HD. In contrast, the DC shared between AD and HD showed a larger proportion of gene pairs that have lost co-expression. This could suggest a greater role for the LOC-related disruptive changes in the pathological mechanisms common to both diseases. Moreover, genes harboring common genetic variants unequivocally found to be associated with AD were more likely to show LOC, despite GOC being a more dominant feature among DC genes in AD. This may indicate that LOC signatures are more likely than GOC genes to be in causal relationship with the onset and/or progression of the disease. LOC interactions of genes at the interface between different DC modules also revealed new candidate disease genes like *FAM59B*. Finally, our systematic search for physical interactions mediating the common DC pairs between AD and HD, besides revealing extensive molecular alterations involving genes such as *GSN* and *SOX10* related to oligodendrocyte differentiation, revealed an interesting split of GOC and LOC dysregulation, respectively, of two physically inter-connected cellular processes, chromatin organization and neural differentiation. Results of our two brain-specific conditional knockout mice validate the interconnection of the two processes as well as key regulator predictions in the subnetwork.

Aging is linked with multiple neurodegenerative diseases including AD and HD. Though most DC patterns identified in the HBTRC data are distinct from age-associated DC pairs derived from the aging study (Colantuoni *et al*, 2011) as seen above, the 242-gene subnetwork dysregulated in both AD and HD overlaps with DE signature sets derived from aging brains. For instance, the 242-gene subnetwork is enriched for genes upregulated in the frontal cortex of old adults (age 73 or more) when compared with younger cortices (age 42 or less) (Lu *et al*, 2004) ($P = 2.38 \times 10^{-39}$), but not for age downregulated genes from the same study ($P = 0.27$). The enrichments are similar with filtered subsets of these signatures reported in the aging study using reversed changes in fetal development as a filter (Supplementary Table S5 of (Colantuoni *et al*, 2011); enrichment $P = 9.6 \times 10^{-9}$ versus $P = 0.39$ for filtered age upregulated versus downregulated set, respectively). Interestingly, most of the age upregulated genes in the 242-gene subnetwork participated in only LOC interactions. Together, these results suggest that aging may contribute to AD and HD risks by increasing the expression of certain genes in the aligned subnetwork whose co-regulation patterns are disrupted in disease.

Comparing molecular and macro-scale networks is not straightforward; however, there are studies of both cell function and macro-scale networks in the human brain which suggest that gain of co-expression (GOC), a dominant feature in our DC networks, may indicate increased functional activity. For instance, (1) Buckner *et al* 2009 used functional neuroimaging (fMRI) to demonstrate that the human cortex contains hubs of high functional connectivity correlating with incidence of Aβ deposition in AD patients, (2) Aβ production is strongly stimulated as a function of increasing neuronal activity (Cirrito *et al*, 2005), and (3) neuronal activity is highly increased (50% of the neuronal population) in the vicinity of Aβ plaques in an early-stage AD mouse model (Busche *et al*, 2008; Kuchibhotla *et al*, 2009), with neuronal hyperactive firing in the cortex combined with an increased astrocyte activity and Aβ plaques. While gain of transcriptional co-regulation in the brain cortex network may be associated with increased activity surrounding misfolded Aβ deposits, it is possible that lack of transcriptional co-regulation (LOC), which is proportionally high in the overlap between AD and HD and high among genes found to be causally related to AD, is associated with upstream events that produce or maintain the misfolded protein aggregates.

Spurious correlations due to systematic noise in the control samples could result in false positive LOC pairs, especially for genes with low expression variation between AD/HD and controls (see also Supplementary Text A.1). However, we showed that the correlation of LOC pairs in our control samples got replicated in an independent cohort's control brain samples. We also showed that genes implicated in AD in independent GWAS studies predominantly participated in LOC pairs and modules of genes comprising several LOC pairs enriched for meaningful biological processes. Furthermore, we verified the robustness of the AD DC network against false positive errors through replication of both GOC/LOC direction and magnitude of Q statistics in the independent human cohort. It is worth noting that exclusion of age-associated DC pairs does not affect the significance of replication of LOC pairs ($P < 1/1,000$ using a permutation test shuffling gene labels), but it increases the replication $P$-value of GOC pairs to $P < 14/1,000$. This result further supports the importance of LOC pairs in disease pathogenesis.

Previous studies on differential co-expresssion analysis in AD have used either module-based (Zhang *et al*, 2013) or hub-based (Rhinn *et al*, 2013) analysis. The module-based analysis defines the co-expression modules in case and control groups separately before comparing their correlation structures between the groups, whereas the hub-based analysis aggregates the differential co-expression of each gene with all other relevant genes in the transcriptome to prioritize disease genes. We take a more direct "edge/pair-based" approach based on meta-analysis of correlation coefficients to identify all pairs of genes exhibiting differential co-expression at 1% FDR. There are several advantages to our edge-based analysis. First, it enables a direct overlap of the DC networks of multiple diseases and alignment of the resulting DC network with other types of networks such as a physical network of protein–protein and protein–DNA interactions. The 242-gene subnetwork resulting from this network alignment revealed common molecular mechanisms underlying AD and HD. Second, our pair-based approach offers a finer resolution of transcriptional dysregulation that allows us to inspect DC patterns not only within modules or hubs, but also at the level of the overall network or at interfaces between two modules. Indeed, we showed that LOC pairs, which are functionally important and robustly replicated in the independent dataset, were overall enriched in the common DC network and arrived at a novel candidate disease association *FAM59B* by inspecting inter-module LOC pairs.

In summary, this study provides a global view of dysregulatory networks in AD and HD through integrative analysis of data from large cohorts of individuals in varying stages of neurodegeneration and aging. While our study falls short of providing a detailed model of disease progression due to the non-longitudinal nature of these datasets, the dysfunctional DC patterns we found in common between AD and HD, and the supporting physical interactions connecting dysregulated molecular pathways (available as Supplementary Datasets D1, D2 and D3) significantly advance current efforts in identifying candidate genes for functional follow-up in independent clinical sampling and drug discovery efforts, which are aimed at influencing and/or modifying susceptibility to both neurodegenerative diseases.

# Materials and Methods

### The human brain samples

The HBTRC (Harvard Brain Tissue Resource Center) samples were primarily of Caucasian ancestry, as only eight non-Caucasian outliers were identified, and therefore excluded for further analysis. Postmortem interval (PMI) was $17.8 \pm 8.3$ h (mean $\pm$ standard deviation), sample pH was $6.4 \pm 0.3$, and RNA integrity number (RIN) was $6.8 \pm 0.8$ for the average sample in the overall cohort. The tissue samples were profiled on a custom-made Agilent 44K array of 40,638 DNA probes uniquely targeting 39,909 mRNA transcripts of 19,198 known and predicted genes (Supplementary Dataset D6). Therefore in some cases, transcripts are targeted by more than one reporter probe, but for ease of notation we refer to these as genes as any duplicate measures are routinely removed during subsequent analyses. After extensive quality control of the samples, 624 DLPFC (BA9) brain tissues from AD patients, HD patients, and non-demented controls ($N = 310$, 157, and 157, respectively) were used for further analysis.

### Adjustment of the gene expression data

As described earlier (Greenawalt *et al*, 2011), we used principal components (PCs) derived from the expression data of the control probes in the Human 44k v1.1 array to adjust the expression data of the other probes, in order to mitigate the effect of unknown confounding factors (Gagnon-Bartsch & Speed, 2012). This adjustment was done in a linear regression setting using the selected PCs as covariates. Following the earlier method (Greenawalt *et al*, 2011), we selected as covariates the 1st PC from r60 control probes, the 1st PC from Pro25G control probes, and the PCs of all remaining control probes that had variance explained at $P < 1e–04$ when compared to a randomized version (obtained from 10,000 permutations of the original data, with each control probe permuted independently). The resulting control-probes-adjusted expression data were further adjusted for several factors that could potentially confound the differences between normal and AD/HD datasets. These adjusted covariates include age, gender, RIN, pH, PMI, batch, and preservation of the samples. Specifically, a robust linear regression model of each gene's expression data was fitted separately for the AD, HD, and control group using these covariates, and residues from the fitted models were taken as the adjusted data for further analyses (rlm in R library MASS with Huber bisquare proposal was used to fit the model). The missing expression values in all expression datasets used in this study, including external datasets, were imputed using the $k$-nearest neighbors algorithm in the space of genes with $k = 10$ (Troyanskaya *et al*, 2001).

### Genotyping and association testing of disease SNPs

Each subject was genotyped on two different platforms, the Illumina HumanHap650Y array (IL650) and a custom Perlegen 300K array (a focused panel for detection of singleton SNPs; PL300). Counting only the union of markers from both genotyping platforms (114,925 SNPs were in the intersection), a total of 838,958 unique SNPs were used for analysis. Restriction Fragment Length Polymorphism (RFLP) was used to genotype the *APOE* polymorphisms in the HBTRC samples as described earlier (Gioia *et al*, 1998). Briefly, the initial PCR yields an amplicon of 485 bp of *APOE* exon 4 containing both polymorphisms, following a nested PCR product of 300 bp. *Hha*I digestion of the nested amplicon generated unique patterns of restriction fragments depending on the original genotype of the individual (Gioia *et al*, 1998). The *GAB2* SNP rs2373115 (Reiman *et al*, 2007) was not present on the arrays and was therefore genotyped using a TaqMan assay. The single point association testing of *APOE* genotypes and other SNPs (reported in Supplementary Table S2) was carried out in the R statistical environment using a logistic regression model encoded by the R formula: disease_status ~ logistic ($\beta0 + \beta1 \times$ (count of minor allele) $+ \beta2 \times$ age $+ \beta3 \times$ gender). The disease SNP $P$-value indicates how significantly different $\beta1$ is from 0. The OR estimate was calculated from $\exp(\beta1)$. We applied the following quality control filters to retain only SNPs that have MAF (Minor Allele Frequency) $> 0.05$, HWE (Hardy–Weinberg Equilibrium) test $P$-value $> 10e–06$, and SNP call rate $> 0.90$.

## Meta-analysis of gene–gene correlation

We used a parametric meta-analysis method to test for changes in gene–gene correlation between two groups (e.g., disease versus controls), under the assumption that gene pairs are bivariate normally distributed in each group. This method yielded similar results as (and somewhat more conservative *P*-values than) a bootstrap method with no parametric assumptions, on a random subset of gene pairs in our data (see Supplementary Fig S7 for these results and a description of the bootstrap method, which is computationally intensive even on a subset of all gene pairs). We now describe the parametric method: for each gene pair ($i,j$) and their Spearman correlation coefficients $r_{tij}$ (with $t = 1$ and 2 computed in disease and control samples, respectively), we first transformed the correlation coefficients into Fisher's *Z*-statistics $z_{tij} = \frac{1}{2}\log\left(\frac{1+r_{tij}}{1-r_{tij}}\right)$, which follows a normal distribution with zero mean and standard deviation of $\frac{1}{\sqrt{n_t-3}}$ ($n_t$ is the sample size) under our parametric assumption of bivariate normality. A heterogeneity statistic $Q$ is then computed for each gene pair as shown next (without the i,j subscripts for clarity): $Q = \sum_t w_t(z_t - \bar{z})^2$ with the weights $w_t = n_t - 3$ being used to also compute $\bar{z}$ as the weighted average of the $z$ in the disease and the control group. The $Q$ statistic follows a $\chi^2$ distribution with one degree of freedom under homogeneity and parametric assumptions (Hedges & Olkin, 1985), and the larger it is, the less similar the gene–gene correlation is between the two groups. To make differential co-expression calls from the $Q$ statistics of all gene pairs taken together, we used a global permutation-based approach that both accounts for multiple testing and is robust to any violations of parametric assumptions (Storey & Tibshirani, 2003). We specifically permuted the original data once (by randomly assigning sample labels to shuffle the two groups together) and repeated the meta-analysis procedure. Then for any global cutoff value $Q_0$, we take the ratio of the number of gene pairs that had $Q > Q_0$ in the permuted to the original data as the estimated global FDR (false discovery rate) at this cutoff. The final cutoff we chose corresponding to FDR 1% translates to: $Q_0 = 25.6$ for AD versus controls comparison, and $Q_0 = 21.7$ for HD versus controls comparison. Unless specified otherwise, in addition to requiring a gene pair to have $Q > Q_0$, the pair has to be significantly co-expressed in either the disease or the control group of samples (but not both) to be called as a differentially co-expressed or DC pair. The GOC and LOC category of DC pairs is defined based on which group the gene pair is significantly co-expressed. We call a pair as significantly co-expressed in a group of samples if their Spearman's correlation test *P*-value is at most 0.01 after correction for the all reporter–reporter tests among the 40,638 reporters, and not significantly co-expressed otherwise.

## Independent human cohorts used in replication testing

We tested replication of the DC pairs identified using the HBTRC samples in an independent human cohort of late-onset AD and control individuals (Webster *et al*, 2009). We obtained the expression data of that study from GEO (GSE15222), extracted the data of postmortem frontal cortex samples alone of 31 AD and 40 control individuals (over 24,354 transcripts, which became 23,613 unique transcripts after replacing transcripts represented by multiple probes with a randomly pre-selected probe), and adjusted through a linear regression model the AD and control group data separately for the same set of covariates used in the study. To dissect the contribution of aging to differential co-expression, we used another independent human dataset comprising expression data of neuropathology-free postmortem prefrontal cortex samples (Colantuoni *et al*, 2011) of 56 elder (age between 50 and 90 at time of death) and 53 adult (age between 20 and 40 at time of death) individuals. We obtained the preprocessed expression data of that study from GEO (GSE30272) before SVA (Surrogate Variable Analysis) adjustment, extracted the data of the elder and adult group of individuals alone, and adjusted through a linear regression model each group's data separately for all non-SVA covariates reported in the study (with the exception of 'smoking history' as this covariate was highly correlated to and had more missing data than the 'smoke at death' covariate).

## Identification of modules in the DC transcriptional network

We used our previously published clustering method described in detail in Wang *et al* (2009). Briefly, the method uses spectral techniques to derive a clustering tree from the DC network obtained from AD versus controls or HD versus controls comparison, and modularity score as an objective function to parse the clustering tree into modules or clusters (of size at least 10 and at most 100) that contain more DC interactions than expected from a random model. The modules were numbered based on their modularity scores, with M1 being the module with the highest modularity score in the network, M2 being the next highest, and so forth. The enrichment *P*-values used for pathway enrichment were calculated using a hypergeometric distribution. All DC modules were tested for enrichment of functional annotations and all significant enrichments ($P \leq 0.05$ after Bonferroni correction for the number of Ingenuity Pathways tested) are reported (but for a module showing multiple significant enrichments, only the best one is reported).

## Alignment of physical interaction network and common DC network

Since large-scale collections of published physical (protein–protein and protein–DNA) interactions are not yet sufficiently comprehensive (Mathivanan *et al*, 2006), we aligned the physical network (viewed as undirected network after dropping edge orientations) with the network of common DC pairs identified in both diseases using a method that is more robust than simply overlapping the edges in the two networks. This alignment method specifically sets out to find all maximal subsets of genes that are connected in both the physical network and the common DC network. Maximal subsets of genes that are connected in two given networks can be found recursively using a simplified variant of a provably efficient algorithm, Match-and-Split, which we have developed previously (Narayanan & Karp, 2007). This variant would find all connected components in the first network, and for each such component, output them if they are also connected in the second network or split them into further connected components in the second network otherwise. This process is repeated recursively until all components of a certain minimum size (10 genes) that are connected in both networks are found.

### *Dnmt1* and *Dnmt3a* brain-specific conditional knockout experiments

All of the mice used in this study were handled in accordance with IACUC-approved protocols. *Dnmt1$^{flox/flox}$* (Fan *et al*, 2001; Jackson-Grusby *et al*, 2001) and *Dnmt3a$^{flox/flox}$* (Nguyen *et al*, 2007) mice were backcrossed onto a C57BL/6 background and crossed with *Olig1-cre* mice to generate *Dnmt1* conditional knockout (*Olig1$^{cre/+}$*; *Dnmt1$^{flox/flox}$*) and littermate control (*Olig1$^{+/+}$;Dnmt1$^{flox/flox}$*) mice, and *Dnmt3a* conditional knockout (*Olig1$^{cre/+}$;Dnmt3a$^{flox/flox}$*) and littermate control (*Olig1$^{+/+}$;Dnmt3a$^{flox/flox}$*) mice. Cortices were dissected from *Dnmt1* conditional knockout (CKO), *Dnmt3a* CKO, and respective littermate control mice at postnatal day 16. *Dnmt1* and *Dnmt3a* were knocked out in oligodendrocytes specifically (Supplementary Fig S8) with recombination rate over 70% (Zhang *et al*, 2009). RNA was isolated from three biological replicates for each genotype using TRIzol (Invitrogen) extraction and isopropanol precipitation. RNA samples were resuspended in water and further purified with RNeasy columns with on-column DNase treatment (Qiagen). RNA purity was assessed by measuring the A260/A280 ratio using a NanoDrop and RNA quality checked using an Agilent 2100 Bioanalyzer (Agilent Technologies). Approximately 250 ng of total RNA per sample were used for library construction by the TruSeq RNA Sample Prep Kit (Illumina) and sequenced using the Illumina HiSeq 2500 instrument according to the manufacturer's instructions. Sequence reads were aligned to the mouse genome (assembly mm10) and gene expression quantified using Tophat/Cufflinks methods, and differentially expressed genes at FDR 10% ($Q$-value < 0.1) were identified using Cuffdiff (Trapnell *et al*, 2013). To compare gene signatures of mouse experiments with human networks, we used human–mouse gene orthologs provided by The Jackson Laboratory (http://www.informatics.jax.org/homology.shtml).

### Data availability

Gene expression data of the human brain samples used in this study are available at the GEO public database under the accession number GSE33000.

Gene expression data from the mouse conditional knockout experiments are also publicly available under the GEO accession number GSE58261.

**Supplementary information for** this article is available online: http://msb.embopress.org

### Acknowledgements

This research was supported in part by NIH Award R01AG046170 (to BZ, ES and JZ), NIH-NIMH (R01MH090948-01 to JZ), NIH-NINDS (R37NS042925-10 to PC and F31NS077504 to JLH), and the Intramural Research Program of the NIH (specifically NIAID and NHLBI institutes for MN and AJ, respectively). [Correction added after first online publication on 30 July 2014: in the preceeding sentence "NIH Award R01AG046170 (to BZ, ES and JZ)," was added] We thank the Harvard Brain Tissue Resource Center (which was supported in part by PHS Grant R24 MH068855, http://www.brain-bank.mclean.org/) for generously gifting human brain postmortem samples used in this study. We thank John Tsang for useful discussions related to enrichment analysis, Carlo Colantuoni for timely access to the adjusted expression dataset described in his publication, and Yuri Kotliarov, Bhaskar Dutta and Zhi Xie for helpful discussions. The network figures were generated using the Cytoscape software (www.cytoscape.org).

### Author contributions

MN, VE, and JZ designed the data analysis study. MN and JZ developed analytical methods and carried out the analyses. MN, VE, and JZ drafted the manuscript. JLH performed the conditional knockout experiments and SY analyzed the RNA-seq data. KW, JM, BZ, CZ, JRL, TX, CS, CM, SM, ADJ, GF, DJS, and EES contributed to and aided the data analysis. XY and PC aided in analyzing, interpreting, and summarizing results. All authors were involved in discussing the results and commented on the manuscript.

### Conflict of interest

Some authors own stocks in their respective companies.

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
