## [Review Process File · Molecular Systems Biology]

Common Dysregulation Network in the Human Prefrontal Cortex underlies Two Neurodegenerative Diseases

Manikandan Narayanan, Kai Wang, Xia Yang, Albert V. Smith, Bin Zhang, Joshua McElwee, Chunsheng Zhang, Sigurdur Sigurdsson, John Lamb, Tao Xie, Christine Suver, Oscar Puig, Cliona Molony, Mr. Hong Bao, Stacey Melquist, Andrew D. Johnson, Lenore J. Launer, Adron Harris, R Dayne Mayfield, David Stone, Vilmundur Gudnason, Eric Schadt, Prof. Jun Zhu

Corresponding authors: Manikandan Narayanan, National Institute of Allergy and Infectious Diseases, National Institutes of Health and Jun Zhu, Icahn School of Medicine at Mount Sinai, NY

Review timeline:

(Submission of initial manuscript:	17 October 2011
Editorial Decision:	07 December 2011)
New submission:	23 December 2013
Editorial Decision:	17 March 2014
Re-submission:	21 March 2014
Editorial Decision:	11 May 2014
Revision received:	11 June 2014
Accepted:	20 June 2014

Editor: Andrew Hufton/Thomas Lemberger

Transaction Report:

1st Editorial Decision

07 December 2011

Thank you again for submitting your work to Molecular Systems Biology. We have now heard back from two of the three reviewers who agreed to evaluate your manuscript. As you will see from the reports below, the referees raise substantial concerns on your work, which, I am afraid to say, preclude its publication in Molecular Systems Biology.

Both reviewers acknowledged the substantial amount of new data presented in this work; however, both independently indicated that were not convinced that this work offered the kind of depth of biological insight or conclusiveness that would be expected in Molecular Systems Biology. The reviewers found the novel conceptual insights into the molecular systems specifically underlying Alzheimer's or Huntington's disease to be modest, and indicated that these data deserved deeper functional investigation and more rigorous validation. In addition, both reviewers had clear concerns regarding the conclusiveness of the mouse model expression profiling results. The second reviewer also raised important technical concerns regarding the statistical analyses, which s/he felt were sufficient to case doubt on some of the results of this study.

Given these concerns, and the low-level of support from the reviewers, I see no other choice than to return the manuscript with the message that we cannot offer to publish it.

In any case, thank you for the opportunity to examine your work. I hope that the points raised in the reports will prove useful to you and that you will not be discouraged from submitting future work to Molecular Systems Biology.

Reviewer #2

"Disrupted patterns of co-regulation in two neurodegenerative diseases" by Narayanan et al. In this manuscript the authors investigate genome-wide disruptions in the co-regulation of genes in Alzheimer's and Huntington's diseases. They analyzed the gene expression profiles of 624 postmortem prefrontal cortex samples and correlated this data with protein-protein and protein-DNA interaction networks. The data presented in the study is indeed very interesting and valuable but, in my opinion, the analyses included in the manuscript and their interpretation are quite shallow. Thus, unfortunately, I cannot recommend its publication in Mol Syst Biol in its present form. These are very good groups, I'm sure they can improve the analyses presented and exploit the data in full.

As I said, I think that the data generated is very interesting and offers a plethora of possibilities for further analyses. However, after having read the manuscript a couple of times, I still haven't found a clear take-home message. The first finding is that a gain of co-regulation (GOC) in AD and HD is more frequent than a loss of co-regulation (LOC), however, when the co-regulated pairs for the two diseases are compared, it seems that the majority of the common pairs show a LOC. From this point on, the authors mainly run some standard function annotation procedures (i.e. clustering, functional enrichment, etc) for all the co-regulated pairs and find some very obvious or general terms (i.e. neurotransmission-related processes, protein folding or metabolic processes). They present these results in a purely descriptive way that doesn't add much to the current knowledge. Given that they see clear differences, in terms of GOC and LOC frequencies, in the total of co-regulated pairs and those common to AD and HD, it'd be interesting to see, for instance, if the functions/processes that are enriched change between these two sets, and then speculate about the meaning of these differences, if any (i.e. are they related to the cause of the diseases or a mere consequence of the brain being damaged).

Almost the same comments apply to the "functional organization of the DC network in AD and HD brains", where only a description of the bioinformatics enrichment analysis is provided. It'd be good to dig deeper into a couple of examples to illustrate the possibilities of the analysis and hypothesize about the modules enriched for genes that correlate with the Braak stages of the diseases. The integration with protein interaction networks is also interesting but, again, no hypotheses are provided. In addition, more information should be given on the network alignment method and the results that it yielded since, looking at Figure 4, it seems that the initial networks are pretty large, and the aligned part is ridiculously small. What does it mean? is it because a small number of common nodes or is it the alignment algorithm that is filtering out other solutions? In any case, I guess that the presented module is not the only solution.

I also did not fully understand the validation using a mouse model and, again, the text is too vague. If I got it right, the authors claim "consistency" between the set of co-regulated genes found in mice and in human, but the similarity, when compared to the background, is insignificant.

Finally, I think that some of the figures do not add much to the manuscript. For instance, Figures 2 and 3 show some protein modules, but no explanation is given in the text as to what they might represent (apart from saying that they are modules). An almost the same applies to Figure 4, where the network around GSN is presented, but no mechanistic hypothesis span from it.

Overall, and given the low level of details given, I have the impression that the manuscript is about AD and HD only because the initial data come from AD and HD brains, but almost the same article could be written by any other set of DC network. As I said repeatedly, I believe that the data presented is very valuable, but the authors should exploit it better and try to add value to the current knowledge of AD and HD which, unfortunately, is not the case with the present form of the manuscript.

Reviewer #3:

This is an ambitious study of two neurodegenerative diseases. A remarkably large number of samples were used and some results seem fairly incontrovertible.

There appears to be a bias towards gain of correlation (GOC) in the co-expression network of both neurodegenerative diseases. This is not that surprising because this result is reproduced in many other disease systems (e.g. cancer) which are also characterized by GOC presumably due to the loss of the homeostatic stochasticity characteristic. The insight that the smaller number of loss-of-correlation (LOC) pairs in the co-expression network reflects more of a disease specific effect is intriguing but not convincingly supported by the data. The shared expression differentially correlated (DC) pairs in HD and AD is interesting but it is not clear from this study whether such a shared DC would be seen with most cerebral diseases, with all neurodegenerative diseases (including microvascular) or just AD and HD. The analysis showing shared signatures with diabetes mellitus is tangential and without performing the same analysis for other non-CNS diseases, the overlap found might not provide any particular insight except that many aging-associated processes in many tissues are shared. That would not be a novel insight.

The finding that the hub genes are enriched for neurotransmission, protein folding and development is also interesting but there are many reasons why this might be so and the fact that they are shared (more than expected by chance) across HD and AD again might say something more about the aging process in neural tissue rather than anything specific to the HD and AD pathologies.

The mouse model results are not particularly illuminating as the authors themselves appear to acknowledge.

The replication of the DC set in another AD population is the most reassuring finding

Also there were several methodological or manuscript problems that detracted from the study:

-They offer no justification for their PCA-based preprocessing adjustment of the expression data. If it's a novel approach, we would need an explanation of what they're trying to achieve, and a statistical analysis of how it performs. If it's an off-the-shelf tool, we need a citation at least, and should also get a justification of why this method over any other.

"we first transformed the correlation coefficients into Fisher's Z-statistics ... which follows a normal distribution with zero mean and standard deviation of ...": This assumption is only true under the following conditions: Only if the variables over which they're computing correlation (X and Y) are bivariate normally distributed $\sim N(X, Y)$. They never tested that assumption, and if not true, this breaks:

- "The Q statistic should follow a χ^2 distribution with one degree of freedom under the homogeneity assumption..." again, if the assumption of normality is broken, then this is not necessarily true.

-They seem to acknowledge the failure of this assumption, though not explicitly: instead of using chi-squared table, they perform permutation simulations (i.e., they probably looked at the results of using a parametric threshold and didn't like what they saw). I think we're all in agreement with the fact that biology is not easily parameterized, but the way the analysis is written suggests that the authors are not being fully transparent.

-With regard to the permutation test, details are missing. They don't report how many permutations they ran, nor exactly how they did the simulation. It's not clear whether they did independent permutations for each gene pair, or whether ran one simulation and picked a global threshold.

-Also, they make no mention of multiple hypothesis testing. If there are N genes, they are trying to reject H_0 $O(N^2)$ times (since there are $O(N^2)$ gene pairs). It seems likely that they ran the right simulation to generate the null distribution they want (perhaps the score threshold changes for each gene), However, in the absence of accounting for the N^2 . multiple hypothesis testing, the 1% FDR

estimate from their simulation is not well supported.

Many of the methodological problem here stem from the fact that they've added the extra layer of the homogeneity score. The classic solution to this type of "are my correlation coefficients statistically different from yours" is a bootstrap procedure. In this instance, one would resample (for each gene) the patients in each disease /phenotype category, say 1000 times. That allows one to estimate your confidence in the observed sample correlation for the two populations. Once you have those, it allows you one to directly assess the probability of seeing phenotype A's sample correlation given phenotype B's bootstrapped distribution, and vice versa. Then you can easily do Bonferroni (or similar) on those.

Re-submission

23 December 2013

Please find enclosed a resubmission of our manuscript "Common Dysregulation Network in the Human Prefrontal Cortex underlies Two Neurodegenerative Diseases", which we would like to be considered for publication as a research article in *Molecular Systems Biology*. In this manuscript, we present global insights into shared dysregulation patterns induced by two neurodegenerative diseases, Alzheimer's and Huntington's disease, as derived from systematic analysis of expression and clinical data from 624 postmortem brain samples. The brain samples in this study constitute one of the largest expression cohorts for AD and to the best of our knowledge the largest expression cohort for HD (covering almost 1% of all HD cases in the US), and the findings in this study are derived from systematically integrating our large-scale data with independent molecular interaction data and independent human cohorts on neurodegeneration and aging. We hence believe the scope of the data and novelty of the integrative research presented here is of deep relevance to understanding the common mechanisms of neurodegeneration and would be of high interest to your journal's readership, especially readers oriented towards understanding system-wide biological regulation through integrative genomic analysis.

In fact our enthusiasm about the scope of the data and replication of findings in an independent human cohort were shared by Editor Dr. Andrew Hufton and reviewers during our earlier submission. They had also raised certain criticisms and concerns, which we have now addressed substantially in the significantly revised manuscript attached herewith. As detailed in the eResponse to reviewers' document, the revised manuscript offers a much improved presentation by augmenting each data analysis with a specific hypotheses of known and novel genes/pathways/interactions involved in neurodegeneration, demonstrating that DC results replicate in independent human cohorts even after accounting for age-related factors and clarifying methodological details to show that our statistical method is sound and achieves similar results to an alternate method suggested by a reviewer.

There are currently no effective treatments that can prevent or halt progression of Alzheimer's and Huntington diseases, providing a strong incentive for development of new research strategies to identify pathways that cause or associate with these diseases. Through a rigorous analysis of our high-dimensional dataset, we demonstrate striking changes in the transcriptional co-ordination of genes and gene modules in the neurodegenerative human brain and many commonalities between the two diseases. Interestingly, the network of common disruptions shows distinctive features related to loss of co-regulation that is not apparent when investigating each disease separately – these findings would hence be missed by current expression studies on individual diseases such as Alzheimer's disease alone. Furthermore, integrating this common network with large-scale physical (protein or regulatory) interaction data led to two physically interacting biological processes with surprisingly dichotomous patterns of disease-induced disruptions (viz., loss of co-regulation in the neuronal differentiation process and gain of co-regulation in chromatin organization). The aggregate molecular disruptions were replicated using independent human cohorts on neurodegeneration and aging. Overall, our work comprehensively uncovers the molecular interactions affected by dementia, and links neurodegeneration to the derailment of specific genes and pathways and the protein-protein/protein-DNA interactions that could propagate these disruptions. As such, our work complements well and addresses many of the limitations of current (yet successful) genome-wide association studies (GWAS) and global expression studies that focus exclusively on one neurodegenerative disease alone.

2nd Editorial Decision

16 January 2014

Thank you for having submitted a manuscript entitled "Common Dysregulation Network in the Human Prefrontal Cortex underlies Two Neurodegenerative Diseases" for consideration for publication in *Molecular Systems Biology*.

First of all, I would like to greatly apologize for the exceptional delay in getting back to you which was due to the Christmas break.

As you may know, our editorial policy does not allow to re-consider a manuscript that was rejected before unless this course of action has been explicitly encouraged in our decision letter. I have nevertheless have now had the chance to read your study and reply letter. We recognize that you have clarified some aspects of the study as compared to the manuscript submitted before. We also acknowledge that you have added some potentially intriguing observations, such as the LOC/GOC dichotomy between the neural differentiation-related genes and chromatin organization genes of the AD/HD common DC gene network. The replication data, as was already included in the previous versions, remains encouraging. We are however not convinced that the insights resulting from this analysis have very significantly changed or progressed as compared to the study evaluated two years ago and some of the more fundamental concerns raised at that time remain. As such, I am afraid that we are not convinced that we should reverse our previous decision and I see no other choice than to return the study with the message that we cannot offer to publish it.

I am very sorry not to be able to bring better news on this occasion and apologize again for the slow process.

Appeal

12 February 2014

The comments from previous reviewers are encouraging and constructive. We have significantly revised our manuscript according to reviewer suggestions and add many new results. We fully respect your assessment of our revised manuscript, but respectfully disagree. We were shocked to hear your conclusion that there is no new insight derived from our study.

First, we validated our differential connectivity (DC) patterns in multiple independent human brain data sets. We showed that DC patterns reflect aging process. Importantly, when removing natural aging related patterns, our DC patterns are even more significantly replicated in independent AD data set, which suggesting DC patterns truly capture pathogenesis of the disease. No other group has ever shown the robustness of their brain related patterns in independent data sets.

Second, we identified a dichotomy subnetwork consisting of interconnected biological processes neural differentiation and chromatin organization that are underlying common pathogenesis of AD and HD. Recent studies suggest there are common mechanisms underlying all neurodegenerative diseases, and hypothesize immune response as the common mechanism. Our data driven approach identified neural differentiation and chromatin organization as the common mechanisms. This finding itself is novel.

Third, we explicitly validated the interconnection of neural differentiation and chromatin organization in the dichotomy subnetwork and showed that predicted key regulators and non key regulators contribute differently to the subnetwork. In the dichotomy subnetwork that we identified, there are two methyltransferases, DNMT1 and DNMT3A (attached figure). Our network suggests that DNMT1 is a key regulator (a large node means that it regulates many genes) while DNMT3A is not (a small node). To validate our network predictions, we generated two condition knockout mice. *Dnmt1* flox/flox and *Dnmt3a* flox/flox mice were backcrossed onto a C57BL/6 background and crossed with *Olig1*-cre mice to generate *Dnmt1* conditional knockout (*Olig1*cre/+;*Dnmt1*flox/flox) and littermate control (*Olig1*+/+;*Dnmt1*flox/flox) mice, and *Dnmt3a* conditional knockout (*Olig1*cre/+;*Dnmt3a*flox/flox) and littermate control (*Olig1*+/+;*Dnmt3a*flox/flox) mice. Cortices were dissected from *Dnmt1* conditional knockout (CKO), *Dnmt3a* CKO, and respective littermate control mice. RNA was isolated from three biological replicates for each genotype and were profiled

using RNAseq technology, then quantified by Cufflink. When compared CKO with wildtype mice, Dnmt1 ko signature significantly overlaps with the subnetwork ($p=5.2e-6$). In contrast, Dnmt3a ko signature do not overlap with the subnetwork ($p=0.07$). These results not only validate the interconnection of the two biological processes, but also validate our prediction of key regulators.

With all these results, we think our method and novel biological insights derived from the method will be interesting to broad audience of Molecular Systems Biology. Please let us know whether you would consider it for further review.

Response

14 February 2014

Thank you for your letter on our decision with regard to your manuscript above. We have now had a chance to read your letter and your manuscript again.

We are sorry to hear that you were 'shocked' by our decision and we would thus like to briefly address the three points made in your letter:

- As we indicated in our letter, we do appreciate that you replicate some of your findings in a second AD cohort, which was actually already included in the first submission, two years ago. In particular, you replicate the finding that there is a larger number of GOC than LOC pairs and you reproduce 5% of the DC pairs (even though we note this is obtained by comparing approx. 40'000 HBTRC DC pairs to >400 times more DC pairs in the new cohort). Validation in terms of functional enrichments and replication of the dichotomy subnetworks are however not reported.

- The dichotomy subnetwork is intriguing. The causes and consequences of the corresponding functional alterations -- neuronal+oligodendrocyte differentiation vs chromatin structure - remain however hypothetical in the submitted manuscript.

- The experiments using oligodendrocyte-specific Cre DNMT3A and DNMT1 floxed mice could potentially be interesting. Unless we missed an aspect of the study, it appears that these data were actually *NOT* included in the submitted manuscript MSB-13-5062 (attached for your reference).

As such, we do not feel that we can revert our decision on the submitted manuscript.

If you wish to send a presubmission inquiry with an extended manuscript that would address the points above and explicitly include all the data described in your letter, we would nevertheless be ready to provide you with a preliminary assessment on whether we would send it out for in-depth review or not.

Additional correspondence

17 February 2014

Thank you for your quick response and insightful comments.

For the specific points you raised,

(1) sample sizes of independent cohorts are much smaller than that of AD, HD, and non-dementia samples in our study. As a result, only a few pairs are differentially co-expressed at genome-wide significant level in the independent cohorts and we can't overlap them with DC pairs in HBTRC directly. Instead, we designed several tests to show trends in the independent cohorts for the 28,000 DC pairs identified in HBTRC. Due to platform differences, only 3,569 out of ~28,000 total DC pairs are testable in the independent data set. There are 401 DC pairs in the dichotomy subnetwork. Among them, 50 DC pairs are testable. There are not enough pairs to make statistical comparison in the permutation tests so that we did not report replication results specific to the dichotomy subnetwork.

(2) we agree that the interconnection of neuron+oligodendrocyte differentiation and chromatin structure is still hypothetical. Our previous works (mainly Dr. Casaccia's works) show that histone

deacetylation controls oligodendrocyte differentiation (Shen et al., Nature Neurosciences, 2008), and YY1 (a node in the subnetwork) forms complexes with HDAC1 and HDAC2 modulating oligodendrocyte differentiation (He et al, Nature Neurosciences, 2010). There are suggestive evidences supporting that DNA methylation regulates neuron differentiation (Kim et al, Human Molecular Genetics, 2014, Wu et al. Science, 2010; Wu et al, J Neur Res., 2012). Actually, their results suggest DNMT3A play a key regulatory role in neuron differentiation, different from our prediction based on the subnetwork. That is one of reasons for our pursuing the specific experimental validation of DNMT1.

(3) The full experiment related to DNMT1 and DNMT3A is included in the updated manuscript (attached).

Response (editor)

17 February 2014

Thank you for your clarification. We would be grateful if you could comment on these two points:

- The characterization of the two Olig1-Cre Dnmt1 and Dnmt3a CKO lines is unclear. What is the pattern of recombination in the two Olig1-Cre Dnmt1 and Dnmt3a CKO lines (eg immunohistological data or in situ etc...) and has the loss of both gene products been verified (this seems particularly important given the modest changes in the Dnmt3a mice)?

- Differential gene expression in the mouse models was determined using a t-test p-value <0.05. How many genes are differentially expressed at FDR<10% taking into account for multiple testing? Is the overlap with the dichotomy network still observed?

Additional correspondence

01 March 2014

For your first question, my collaborators put together a set of confocal images of cells stained with lineage specific antibodies and antibodies for Dnmt1 and Dnmt3a. Green: cell markers stains and red: Dnmt1 or Dnmt3a stains. Dnmt1 and Dnmt3a were efficiently knocked out in oligodendrocyte cells. The white arrows indicate the absence of the enzymes only in the nucleus of oligodendrocytes (Olig2+) but not in the nucleus of astrocytes (GFAP+) or neurons (NeuN+).

For your second question, we analyzed the RNAseq data in two ways. We applied the t-test to the FPKM data outputted from Cufflink and used p-value cutoffs to define differentially expressed genes. We also used the CuffDiff program in the Cufflink package to identify differentially expressed genes and used q-value cutoffs to define signature genes. The results are summarized in the excel file. Dnmt1 KO signature genes defined by both methods robustly overlap with our subnetwork significantly and CuffDiff results overlap with our subnetwork more significantly. In contrast, Dnmt3a KO signatures do not overlap well with our subnetwork. We note that the Dnmt1 ko signatures defined by t-test at p-value <0.01 and defined by CuffDiff at q-value<0.1 are similar.

Our enrichment result is not cutoff sensitive.

At p<0.001, there are 88 differentially expressed (DE) genes, 11 of them are in our subnetwork, enrichment p-value = 3.3e-8;

At p<0.005, there are 278 DE genes, 19 of them are in our subnetwork, enrichment p-value = 1.1e-8;

At p<0.01, there are 452 DE genes, 23 of them are in our subnetwork, enrichment p-value = 7.9e-8

At p<0.05, there are 1490 DE genes, 42 of them are in our subnetwork, enrichment p-value = 5.2e-6.

Note that, the hub gene GSN and its neighbor genes (MAG, GJB1, MYO1D) are differentially expressed at p<0.001. SOX10 and ERBB3 are differentially expressed at p<0.005.

When using FDR q value to filter t-test result, there are only 30 or so DE genes at q<0.1.

Please let us know if there is anything you need to evaluate our manuscript.

Editorial Decision

17 March 2014

The results look promising (the % of oligodendrocyte targeted is difficult to assess, but this is a question reviewers should assess). If you can add these data and the additional statistical analysis in the manuscript, we would be pleased to send it out for in-depth review.

Re-submission

21 March 2014

Detailed responses are provided below each comment in blue font type. All page numbers and other such references given are with respect to the revised manuscript unless otherwise stated.

Reviewer #2

"Disrupted patterns of co-regulation in two neurodegenerative diseases" by Narayanan et al. In this manuscript the authors investigate genome-wide disruptions in the co-regulation of genes in Alzheimer's and Huntington's diseases. They analyzed the gene expression profiles of 624 postmortem prefrontal cortex samples and correlated this data with protein-protein and protein-DNA interaction networks. The data presented in the study is indeed very interesting and valuable but, in my opinion, the analyses included in the manuscript and their interpretation are quite shallow. Thus, unfortunately, I cannot recommend its publication in Mol Syst Biol in its present form. These are very good groups, I'm sure they can improve the analyses presented and exploit the data in full.

We thank the reviewer's encouraging comments. We not only mined the data deeper and generated novel insights of common molecular mechanisms underlying AD and HD, but also experimentally validated them (please see the summary above).

As I said, I think that the data generated is very interesting and offers a plethora of possibilities for further analyses. However, after having read the manuscript a couple of times, I still haven't found a clear take-home message. The first finding is that a gain of co-regulation (GOC) in AD and HD is more frequent than a loss of co-regulation (LOC), however, when the co-regulated pairs for the two diseases are compared, it seems that the majority of the common pairs show a LOC. From this point on, the authors mainly run some standard function annotation procedures (i.e. clustering, functional enrichment, etc) for all the co-regulated pairs and find some very obvious or general terms (i.e. neurotransmission-related processes, protein folding or metabolic processes). They present these results in a purely descriptive way that doesn't add much to the current knowledge. Given that they see clear differences, in terms of GOC and LOC frequencies, in the total of co-regulated pairs and those common to AD and HD, it'd be interesting to see, for instance, if the functions/processes that are enriched change between these two sets, and then speculate about the meaning of these differences, if any (i.e. are they related to the cause of the diseases or a mere consequence of the brain being damaged).

We thank the reviewer's constructive suggestions here and below. Comparing genes involved in LOC and GOC in terms of GWAS SNPs, functions, and replication in independent datasets, suggested that genes involved in LOC are more likely to be causal for neurodegenerative diseases (results summarized in Page 9). We have also pointed out the causal regulators in the subnetwork of common DC relations between AD and HD aligned with physical interactions, using AD GWAS genes and HD RNAi screen hits. In general, we've significantly improved the presentation so that only analyses focused on common pathologies in neurodegeneration are presented, along with specific novel hypothesis or new knowledge about neurodegeneration gleaned from each analysis (as described in response to other comments below).

Almost the same comments apply to the "functional organization of the DC network in AD and HD brains", where only a description of the bioinformatics enrichment analysis is provided. It'd be good to dig deeper into a couple of examples to illustrate the possibilities of the analysis and hypothesize about the modules enriched for genes that correlate with the Braak stages of the diseases.

We have significantly revised this section to focus solely on the shared DC modules in both diseases, and how mining the interface (crosstalk) between these modules leads to specific hypotheses of novel disease genes for experimental validation (*FAM59B* in Page 15).

The integration with protein interaction networks is also interesting but, again, no hypotheses are provided. In addition, more information should be given on the network alignment method and the results that it yielded since, looking at Figure 4, it seems that the initial networks are pretty large, and the aligned part is ridiculously small. What does it mean? is it because a small number of common nodes or is it the alignment algorithm that is filtering out other solutions? In any case, I guess that the presented module is not the only solution.

We provided a more detailed description of the network alignment method used (Pages 30-31). The number of edges that directly overlap between the two networks is small despite the large number of interactions in the physical network – one reason for this could be the incompleteness of the human physical network, especially under physiological conditions similar to human brain function or dysfunction. Another reason could be that not all physical interactions lead to transcriptional co-regulation, as the former is a representation of a more static logic hardwired in the genome whereas the latter arises from a highly condition-dependent execution of the static logic to produce various correlated transcripts. In fact, the overlap identified by our network alignment algorithm captures both direct and indirect overlaps, thereby overcoming the sparse overlap problem.

We've now presented a clear hypothesis that spans from this aligned network, namely the link between loss of co-regulation in neuronal/oligodendrocyte-related differentiation and gain of activity in chromatin organization, underlying multiple neurodegenerative diseases, and experimentally validated the connection of the two biological processes. We have also pointed out the causal regulators in this subnetwork, in terms of AD GWAS genes and HD RNAi screen hits. Please find these results in Pages 16-21.

I also did not fully understand the validation using a mouse model and, again, the text is too vague. If I got it right, the authors claim "consistency" between the set of co-regulated genes found in mice and in human, but the similarity, when compared to the background, is insignificant.

Despite multiple differences between human and mouse brains, and differences in experimental designs as well, it was encouraging to see similar trends in the mouse data. But we believe that the very small sample size of the mouse data could've led to statistical insignificance. We've removed the mouse results section in the revised manuscript, and instead strengthened the replication of our results using newly available human datasets as described in Pages 10-12.

Finally, I think that some of the figures do not add much to the manuscript. For instance, Figures 2 and 3 show some protein modules, but no explanation is given in the text as to what they might represent (apart from saying that they are modules). An almost the same applies to Figure 4, where the network around GSN is presented, but no mechanistic hypothesis span from it.

We have significantly revised all figures, so that they are both informative and support the conclusions made in the text. The text explaining Figure 5 now outlines the specific hypothesis linking multiple dysregulated processes like neuronal/oligodendrocyte differentiation and chromatin organization in neurodegeneration, and points to causal AD or HD genes in the subnetwork.

Overall, and given the low level of details given, I have the impression that the manuscript is about AD and HD only because the initial data come from AD and HD brains, but almost the same article could be written by any other set of DC network. As I said repeatedly, I believe that the data presented is very valuable, but the authors should exploit it better and try to add value to the current knowledge of AD and HD which, unfortunately, is not the case with the present form of the manuscript.

We thank the reviewer again for his/her enthusiasm and constructive suggestions, and the significant revisions outlined above demonstrate how our analyses add new knowledge that is highly relevant and specific to shared dysregulation in AD and HD. For instance, the novel association of *FAM59B* to neurodegeneration; the crosstalk between dysregulated processes, neuronal/oligodendrocyte differentiation and chromatin organization, in the 242-gene aligned subnetwork; and the distribution of causal regulators among LOC interactions and in the aligned subnetwork (based on AD GWAS genes or HD RNAi hits), all provide knowledge on transcriptional dysregulation in complex neurodegenerative diseases that could not be obtained from smaller datasets or standard differential expression analyses.

Reviewer #3

This is an ambitious study of two neurodegenerative diseases. A remarkably large number of samples were used and some results seem fairly incontrovertible.

There appears to be a bias towards gain of correlation (GOC) in the co-expression network of both neurodegenerative diseases. This is not that surprising because this result is reproduced in many other disease systems (e.g, cancer) which are also characterized by GOC presumably due to the loss of the homeostatic stochasticity characteristic. The insight that the smaller number of loss-of-correlation (LOC) pairs in the co-expression network reflects more of a disease specific effect is intriguing but not convincingly supported by the data.

We thank the reviewer's insightful comments and suggestions. As we don't have enough data to directly infer causality to diseases for genes involved in LOC or GOC, we instead compared genes involved in LOC and GOC in terms of GWAS SNPS, functions, and replication in independent datasets, and the results suggest that genes involved in LOC are more likely to be causal for neurodegenerative diseases (results summarized in Page 9).

The shared expression differentially correlated (DC) pairs in HD and AD is interesting but it is not clear from this study whether such a shared DC would be seen with most cerebral diseases, with all neurodegenerative diseases (including microvascular) or just AD and HD.

We further focused on the subnetwork where common DC pairs between AD and HD were aligned with the human physical interaction network, and showed that this subnetwork contains signatures of other neurological diseases such as depression (Page 19).

The analysis showing shared signatures with diabetes mellitus is tangential and without performing the same analysis for other non-CNS diseases, the overlap found might not provide any particular insight except that many aging-associated processes in many tissues are shared. That would not be a novel insight.

We recently identified and experimentally validated that APP in pancreatic islet beta cells is a key causal regulator of insulin secretion (Tu et al. PLoS Genetics, 2012), which provides a mechanistic link between AD and T2D. However, systematic comparison with other non-CNS diseases to enforce our claim on the link between AD and T2D will be beyond the scope of the central message of this paper on common transcriptional mechanisms underlying neurodegeneration. Thus, we've taken the reviewer's suggestion and removed the section on T2D.

The finding that the hub genes are enriched for neurotransmission, protein folding and development is also interesting but there are many reasons why this might be so and the fact that they are shared (more than expected by chance) across HD and AD again might say something more about the aging process in neural tissue rather than anything specific to the HD and AD pathologies.

Again, we thank the reviewer's insightful comments. In our dataset, the average age of HD patients is younger than the average of age of non-demented normal control donors, while the average age of AD patients is older than the control's. If age is the main factor for DC, then we would expect overlapped DC pairs to have different signs in the two diseases (such as LOC pairs in AD would be GOC pairs in HD). However in our result, all overlapped DC pairs between AD and HD have the same trends (such as an LOC pair in AD is always LOC in HD also).

We have also carefully compared our DC pairs with DC pairs identified in an independent aging study, and confirmed that most of our DC pairs are not associated with age (Pages 12-13). Our analysis further clarifies which of the DC relations could be due to normal or accelerated aging and which are reflective of age-independent neurodegeneration, and how the replication of our DC pairs in the independent AD dataset is preserved even after excluding age-related DC pairs (please see updated Figure 2B).

The mouse model results are not particularly illuminating as the authors themselves appear to acknowledge.

Despite multiple differences between human and mouse brains, and differences in experimental designs as well, it was encouraging to see similar trends in the mouse data. But we believe that the very small sample size of the mouse data could've led to statistical insignificance. We've removed the mouse results section in the revised manuscript, and instead strengthened the replication of our results using newly available human datasets.

The replication of the DC set in another AD population is the most reassuring finding.

We further tested the replication of this DC set with and without exclusion of age-related features. Specifically, the DC set continues to replicate in the independent Webster et al. data set, before or after removal of age-related DC pairs found using an independent aging dataset (Figure 2B).

Also there were several methodological or manuscript problems that detracted from the study: We carefully addressed all concerns below. In brief, we've clarified the methodological details to show that our DC identification method is based on statistically sound assumptions and provide supplementary results to show that the bootstrap-based DC identification that the reviewer suggested gives similar results to our method, but is more compute-intensive.

-They offer no justification for their PCA-based preprocessing adjustment of the expression data. If it's a novel approach, we would need an explanation of what they're trying to achieve, and a statistical analysis of how it performs. If it's an off-the-shelf tool, we need a citation at least, and should also get a justification of why this method over any other.

PCA-based method to adjust unknown confounding factors has been increasingly used in preprocessing large expression datasets. We have added the appropriate citations and details in Methods section in Pages 25-26.

"we first transformed the correlation coefficients into Fisher's Z-statistics ... which follows a normal distribution with zero mean and standard deviation of ...": This assumption is only true under the following conditions: Only if the variables over which they're computing correlation (X and Y) are bivariate normally distributed $\sim N(X, Y)$. They never tested that assumption, and if not true, this breaks. "The Q statistic should follow a χ^2 distribution with one degree of freedom under the homogeneity assumption..." again, if the assumption of normality is broken, then this is not necessarily true.

-They seem to acknowledge the failure of this assumption, though not explicitly: instead of using chi-squared table, they perform permutation simulations (i.e., they probably looked at the results of using a parametric threshold and didn't like what they saw). I think we're all in agreement with the fact that biology is not easily parameterized, but the way the analysis is written suggests that the authors are not being fully transparent.

We have added more methodological descriptions to clarify certain assumptions, and show in Figure R1 below that only a small fraction of genes or gene pairs in our data show significant or non-modest deviations from normality or bivariate normality. This observation is further supported by the high consistency we see between our parametric meta-analysis-based Pvalues and the non-parametric bootstrap-based Pvalues suggested by the reviewer as described in the response below. We respectfully disagree with the reviewer's comment about multiple testing corrected Chi-squared-based P-values vs. permutation-based P-values. Multiple testing correction methods like Bonferroni assume all tests are independent, which is not true for most genetic and genomic studies, and a frequent alternative in these studies is the empirical permutation-based False Discover Rate (FDR) procedure (Storey and Tibshirani, PNAS 2003), which has the advantage of controlling the overall false discovery rate without requiring assumptions or preconditions on data dependency (such as whether genes are co-regulated, what percentage of genes are co-regulated, or how co-regulated genes are organized). Thus, we used a permutation-based global FDR cutoff in our study.

-With regard to the permutation test, details are missing. They don't report how many permutations they ran, nor exactly how they did the simulation. It's not clear whether they did independent permutations for each gene pair, or whether ran one simulation and picked a global threshold.

We added detailed descriptions on permutation tests in the Method section on Page 28. We used a global permutation test, which generates $\sim 8 \times 10^8$ Q values, to pick a global threshold corresponding to 1% FDR. This threshold was 25.6 for the AD DC network for instance, and we also confirmed that this threshold yielded similar FDR estimates (0.95% to 1.52%) in 10 other independent global permutation tests.

-Also, they make no mention of multiple hypothesis testing. If there are N genes, they are trying to reject H0 $O(N^2)$ times (since there are $O(N^2)$ gene pairs). It seems likely that they ran the right simulation to generate the null distribution they want (perhaps the score threshold changes for each gene), However, in the absence of accounting for the N^2 . multiple hypothesis testing, the 1% FDR estimate from their simulation is not well supported.

We used a global (not gene specific) cutoff instead of Bonferroni multiple testing procedure as described above, since the various tests are not independent and the global cutoff can control the

FDR among all our discoveries taken together. We've updated the methodological description on Page 28 to make this point clear.

Many of the methodological problem here stem from the fact that they've added the extra layer of the homogeneity score. The classic solution to this type of "are my correlation coefficients statistically different from yours" is a bootstrap procedure. In this instance, one would resample (for each gene) the patients in each disease /phenotype category, say 1000 times. That allows one to estimate your confidence in the observed sample correlation for the two populations. Once you have those, it allows you one to directly assess the probability of seeing phenotype A's sample correlation given phenotype B's bootstrapped distribution, and vice versa. Then you can easily do Bonferroni (or similar) on those.

We agree with the reviewer that the bootstrap procedure is an alternative to the meta-analysis method to test for difference between two correlation coefficients. We applied the bootstrap procedure (Suppl Figure S7) to our data and compared the result with our meta-analysis based results, and found high consistency between the two methods (correlation of $-\log_{10}(\text{p-values})$ is $r=0.92$ shown in Suppl Figure S7). In fact, the meta-analysis method is more conservative as shown in Suppl Figure S7, and is much less time-consuming than the bootstrapped method. So we felt confident using the DC pairs from a more conservative and computationally efficient method.

3rd Editorial Decision

11 May 2014

Thank you again for submitting your work to Molecular Systems Biology. We have now heard back from the three referees who accepted to evaluate the study. As you will see, the referees find the topic of your study of potential interest and are now globally supportive. Reviewer #1, who evaluated the study afresh, raises a series of comments with regard to the clarity of the presentation and we would kindly ask you to address them carefully with appropriate modifications.

We appreciate that you provide the accession number to the human expression data. We would suggest that you add a "Data availability" sub-section at the end of the Materials and Methods section where you list the accession number both of the human and mouse knock out datasets.

Reviewer #1:

In the paper "Common Dysregulation Network in the Human Prefrontal Cortex underlies Two Neurodegenerative Diseases," Narayanan and colleagues present a transcriptome-wide analysis of gene expression patterns in the prefrontal cortex of 624 individuals with Alzheimer's disease or Huntington's disease, or with no disease. Specifically, they identified networks of differentially co-expressed (DC) gene pairs that either gained or lost correlation (GOC, LOC) in disease cases relative to the control group, and used these networks to find common expression patterns in both diseases. They then used these data, as well as a subnetwork highlighted from known physical interactions, to identify novel disease genes for follow-up study. For example, mice with brain-wide knockout of the hub gene Dnmt1 show gene expression changes that significantly overlap with this subnetwork, while Dnmt3a KO mice do not. Their main conclusion is that they "show for the first time that the global pattern of gene-gene co-regulation in the human brain cortex is drastically altered in a shared fashion in neurodegenerative diseases like AD and HD."

There are several aspects of the study that are novel and would be interesting for the AD/HD community, but there are also a few things that I found missing or confusing. On the positive side, the concept of comparing two diseases based on changes in connectivity of gene pairs relative to control is a great idea and was applied very well in this context. It is interesting that disease-specific and disease-common changes in DC seem to be in opposite directions (GOC vs. LOC). Also, increasing evidence is pointing to oligodendrocytes (and other non-neuronal cell types) as playing important roles in neurodegenerative diseases, so the results from figures 5-6 seem very reasonable and could potentially be a useful baseline for future disease study. Finally, the study itself is well designed, which, along with the huge set of 624 samples, will allow many possibilities for further analysis.

I also wanted to point out that the added analysis showing that these results are not due to normal aging is quite important and interesting, as several studies have shown similar differential expression changes between AD and normal aging. It would be interesting to see which of the genes in the identified subnetwork tended to show increased or decreased expression with age in addition to showing DC changes between control and AD/HD. Maybe this would be a way of tying in increased disease vulnerability with age.

The presentation and interpretation of some of the results needs to be improved. In general, I had a hard time understanding exactly how the AD and HD networks were supposed to relate to the AD and HD pathologies. This is partially addressed in the middle of page 4, but it would be useful to spell this out more clearly throughout the paper. I also felt that in many cases the figures did not highlight the important points described in the text, as mentioned in several specific comments below. For example, based on the text on page 15 it seems reasonable that FAM59B is associated with neurodegeneration, but Figure 4A doesn't even mention FAM59B, and Figure 4B mentions FAM59B in the context of correlation with Braak stage. This is the only place in the paper, as far as I can tell, that differential expression is addressed at all, other than in supplemental materials.

In the rebuttal you mention that you have removed the mouse results section, and yet there is still a mouse results section about Dnmt1 KO mice. I would suggest keeping this result in the paper.

Supplemental Figures 1-3, Text A.1: As mentioned, the idea behind differential co-expression is that it can identify complementary, and often more disease-relevant, results as differential expression. It would still be nice to see in the main text whether any of these changes in differential co-expression relate to changes in expression with disease. I realize that this is already discussed in supplemental, but I would guess that many readers not bioinformatically inclined would appreciate a bit more of this explanation. (Please consider this an optional change.)

Figure 2B seems like a somewhat complicated way of showing this result. It might be clearer if only a single Q cutoff were shown per plot? Maybe it would also be useful to also include a line for random networks? I realize that something to this effect is in Supp Fig 6, so please consider this an optional change.

Page 14: "Of all DC pairs in the AD network, 36% were intra-module (both genes in a DC pair within the same module), 56% were inter-module (a DC pair interfacing two modules), and the rest were between genes not belonging to any AD module (these percentages were 21% and 74% respectively for HD)." Why are these important? Are these numbers greater than chance? It seems to me that by using the DC data to generate modules, you would expect a lot of the DC pairs to be intra-module.

Related comment - top of Page 15: "The overall topology of the DC module network in Figure 3 also revealed widespread loss of co-regulation in the crosstalk (inter-module) relationship between shared DC modules, and facilitates hypothesis on regulator genes whose disruption lies at the interface of different modules." Again, if you are defining the modules based on the DC networks it does not seem surprising that there is a low amount of cross-talk. If you are saying that the shared DC modules have less cross talk than the AD- and HD-specific modules, that does not appear to be the case based on the plots in figure 3 (i.e., the number of lines connecting between the shared modules does not appear to be any less). It would be helpful to show this in a more straightforward way if this is an important point.

Figure 3: The legend says that LOC is blue and GOC is red, but it appears from the text that the opposite is true. Please correct.

Figure 4a: It is not clear the point of this figure. Are rows gene-gene pairs? If so, label them (or at least the important ones). Which ones correspond to correlations between FAM59B and other genes? If the point is that a lot of the DC pairs involve FAM59B, show which ones. Is the point that there is more LOC in the cross talk than in the M26 only? Are the bottom rows in the M39-only loss of (positive) correlation or gain of (negative) correlation? M39 looks like it has shared GOC in both HD and AD, but is not listed as significant with respect to HD in Figure 3.

Figure 5a: It seems odd that around half of this physical interaction network would be LOC interactions and the other half would be GOC interactions, and that the physical interactions (black lines) appear to mostly be in the center, connecting the LOC and GOC parts of the graph. Would this suggest that normal interactions are likely not to be LOC or GOC in disease, but that the LOC and GOC networks are deviations from such normal interactions (or something to that effect)? This concern appears to be somewhat addressed on pages 17-19, but it is still a bit unclear, and I think this is an important point.

Related comment - page 19: "All these results suggest that the 242-gene subnetwork involving two interacting biological processes, loss of co-regulation in oligodendrocyte differentiation or myelination, and gain of co-regulation in chromatin organization could underlie multiple neurodegenerative diseases." This is a critical point that is buried at the end of a section several paragraphs long. I would suggest making it more clear early in the section that this is what you are getting at with figure 5A.

Page 17: "In this regard, it is notable that the propagation of DC relations between GSN and functionally related genes includes MAG (myelin-associated glycoprotein), a molecule that is synthesized in myelinated oligodendrocytes and localized at the axonal interface (Arroyo & Scherer, 2000; Trapp et al, 1989); GJB1, the gene encoding for Connexin 32, which also localizes in the myelinated fibers of the central nervous system (Scherer et al, 1995); and SOX10, a key transcriptional regulator of myelination in both central and peripheral nervous systems (Stolt et al, 2002)." Okay, but why is this notable? What is the hypothesis? Are you suggesting common dysregulation of oligodendrocytes in AD and HD involving the gene GSN, or something to that effect?

Figure 6: This does not seem to warrant a figure that is identical to figure 5, with the exception of different colored nodes. Would it be possible to combine figure 5 and 6 in some way (maybe by coloring the text for genes changing in the Dnmt1 CKO mice?) or make Figure 6 supplemental?

Middle of page 22: "...our observation that gain of co-expression (GOC, which may indicate increased activity) is a dominant feature in the DC network." If GOC may indicate increased activity, this would be useful to discuss much earlier in the paper. More generally, throughout the paper it would make the results much clearer if the biological interpretation of network concepts were discussed a bit more.

Reviewer #2:

The authors have done a truly thorough and thoughtful revision based upon the Reviewer's queries. I am quite impressed with the revised manuscript, particularly with the mouse model validation strategy, and am satisfied with virtually all of the responses in the letter and revisions in the text.

Revision received

11 June 2014

We thank the reviewers for their thoughtful feedback and constructive comments based on a thorough reading of our manuscript. We have revised the manuscript to carefully address each of these comments.

We highlight main revisions in the manuscript in yellow color, and provide specific responses below each comment here in blue type. All page numbers and other such references given here are with respect to the revised manuscript unless otherwise stated.

Reviewer #1:

In the paper "Common Dysregulation Network in the Human Prefrontal Cortex underlies Two Neurodegenerative Diseases," Narayanan and colleagues present a transcriptome-wide analysis of gene expression patterns in the prefrontal cortex of 624 individuals with Alzheimer's disease or Huntington's disease, or with no disease. Specifically, they identified networks of differentially co-expressed (DC) gene pairs that either gained or lost correlation (GOC, LOC) in disease cases

relative to the control group, and used these networks to find common expression patterns in both diseases. They then used these data, as well as a subnetwork highlighted from known physical interactions, to identify novel disease genes for follow-up study. For example, mice with brain-wide knockout of the hub gene Dnmt1 show gene expression changes that significantly overlap with this subnetwork, while Dnmt3a KO mice do not. Their main conclusion is that they "show for the first time that the global pattern of gene-gene co-regulation in the human brain cortex is drastically altered in a shared fashion in neurodegenerative diseases like AD and HD."

There are several aspects of the study that are novel and would be interesting for the AD/HD community, but there are also a few things that I found missing or confusing. On the positive side, the concept of comparing two diseases based on changes in connectivity of gene pairs relative to control is a great idea and was applied very well in this context. It is interesting that disease-specific and disease-common changes in DC seem to be in opposite directions (GOC vs. LOC). Also, increasing evidence is pointing to oligodendrocytes (and other non-neuronal cell types) as playing important roles in neurodegenerative diseases, so the results from figures 5-6 seem very reasonable and could potentially be a useful baseline for future disease study. Finally, the study itself is well designed, which, along with the huge set of 624 samples, will allow many possibilities for further analysis.

We thank the reviewer for the encouraging feedback. Gene expression profiling data for all human brain samples used in this study have been deposited in GEO as GSE33000, and will be made publicly available upon acceptance of this manuscript.

I also wanted to point out that the added analysis showing that these results are not due to normal aging is quite important and interesting, as several studies have shown similar differential expression changes between AD and normal aging. It would be interesting to see which of the genes in the identified subnetwork tended to show increased or decreased expression with age in addition to showing DC changes between control and AD/HD. Maybe this would be a way of tying in increased disease vulnerability with age.

We thank the reviewer for this suggestion. The 242-gene subnetwork is enriched for genes found to be up-regulated, but not down-regulated, in the aging cortex in an earlier study and interestingly most of them participate in only LOC interactions. We added a paragraph in the Discussion section related to this finding in Page 23.

The presentation and interpretation of some of the results needs to be improved. In general, I had a hard time understanding exactly how the AD and HD networks were supposed to relate to the AD and HD pathologies. This is partially addressed in the middle of page 4, but it would be useful to spell this out more clearly throughout the paper. I also felt that in many cases the figures did not highlight the important points described in the text, as mentioned in several specific comments below. For example, based on the text on page 15 it seems reasonable that FAM59B is associated with neurodegeneration, but Figure 4A doesn't even mention FAM59B, and Figure 4B mentions FAM59B in the context of correlation with Braak stage. This is the only place in the paper, as far as I can tell, that differential expression is addressed at all, other than in supplemental materials.

Your comments are duly noted, and we've tried to add sentences to improve interpretation of the network concepts (eg. Page 7), split large subsections into two to improve presentation (eg. Page 18), and updated figures to highlight the main points described in the text (eg. Figure 4A now lists all pairs in the crosstalk between AD modules M39 and M26, and Figure 4B now provides more information on each AD as well as control sample).

We would like to elaborate a bit more on two other important points you've raised:

- Relation between networks and pathology: The DC networks capture the overall disease-induced disruption of normal gene-gene co-regulatory relations. Since gene-gene co-regulations in normal (non-demented) brain tissues is a footprint of the various biological processes and cell types that are co-ordinated to maintain a healthy tissue, changes in this footprint in AD or HD provides a global view of the causal and reactive dysregulatory events accompanying these diseases. We've discussed this in Page 7.

- Focus on DC (Differential Co-expression) over DE (Differential Expression) analysis: We've moved a section on the complementary nature of results from DC vs. DE analysis from Supplementary Material to the main text (Page 9). We focused the manuscript primarily on the identified DC gene pairs instead of DE genes, since many studies have already looked at genes differentially expressed in AD and HD or those correlated with clinical measures of disease severity as you've mentioned and only a few have looked at gene-gene relations dysregulated in AD or HD.

In the rebuttal you mention that you have removed the mouse results section, and yet there is still a mouse results section about Dnmt1 KO mice. I would suggest keeping this result in the paper. We're keeping the Dnmt1 KO mice results in the paper. The removed section concerns a different mouse dataset that was used before to replicate the overall human DC patterns – this earlier attempt resulted in non-convincing results due to small sample size of this other mouse dataset and differences between disease manifestations in humans and the mouse model. Since we demonstrate that DC patterns can be replicated robustly in independent human brain data sets, we removed the earlier section on replicating overall DC patterns in mouse data.

Supplemental Figures 1-3, Text A.1: As mentioned, the idea behind differential co-expression is that it can identify complementary, and often more disease-relevant, results as differential expression. It would still be nice to see in the main text whether any of these changes in differential co-expression relate to changes in expression with disease. I realize that this is already discussed in supplemental, but I would guess that many readers not bioinformatically inclined would appreciate a bit more of this explanation. (Please consider this an optional change.)

Again, we thank the reviewer for the suggestion. As discussed above, we summarized results on differential expression analysis in Page 9 before focusing the manuscript on differential co-expression analysis.

Figure 2B seems like a somewhat complicated way of showing this result. It might be clearer if only a single Q cutoff were shown per plot? Maybe it would also be useful to also include a line for random networks? I realize that something to this effect is in Supp Fig 6, so please consider this an optional change.

We've incorporated this useful suggestion to obtain a simplified version of Figure 2B (labeled Figure 2C in the revised text).

Page 14: "Of all DC pairs in the AD network, 36% were intra-module (both genes in a DC pair within the same module), 56% were inter-module (a DC pair interfacing two modules), and the rest were between genes not belonging to any AD module (these percentages were 21% and 74% respectively for HD)." Why are these important? Are these numbers greater than chance? It seems to me that by using the DC data to generate modules, you would expect a lot of the DC pairs to be intra-module.

We included these numbers to illustrate a technical point discussed below; however we now realize it may distract the paper from the biological hypotheses that span from these module-module networks. So we've taken your suggestion and removed it. The technical point is that not all networks exhibit a modular structure, and so it is not apparent *a priori* (before looking at these numbers) to what extent the DC networks could be clustered into well-defined modules.

Related comment - top of Page 15: "The overall topology of the DC module network in Figure 3 also revealed widespread loss of co-regulation in the crosstalk (inter-module) relationship between shared DC modules, and facilitates hypothesis on regulator genes whose disruption lies at the interface of different modules." Again, if you are defining the modules based on the DC networks it does not seem surprising that there is a low amount of cross-talk. If you are saying that the shared DC modules have less cross talk than the AD- and HD-specific modules, that does not appear to be the case based on the plots in figure 3 (i.e., the number of lines connecting between the shared modules does not appear to be any less). It would be helpful to show this in a more straightforward way if this is an important point.

The point we wanted to convey was that inter-module DC connections can lead to novel disease association genes such as FAM59B (Figure 4), and so a method for identifying inter-module DC connections is advantageous over other methods that focus solely on intra-module DC identification (eg. (Zhang et al (2013))). Also, we did not mean to highlight there is a low amount of crosstalk; we simply wanted to convey that LOC relations was found among the discovered inter-module (crosstalk) DC pairs and it led to the FAM59B disease association.

Figure 3: The legend says that LOC is blue and GOC is red, but it appears from the text that the opposite is true. Please correct.

Thank you for locating this discrepancy – we have corrected the figure legend now.

Figure 4a: It is not clear the point of this figure. Are rows gene-gene pairs? If so, label them (or at least the important ones). Which ones correspond to correlations between FAM59B and other genes? If the point is that a lot of the DC pairs involve FAM59B, show which ones. Is the point that there is more LOC in the cross talk than in the M26 only? Are the bottom rows in the M39-only loss of (positive) correlation or gain of (negative) correlation? M39 looks like it has shared GOC in both HD and AD, but is not listed as significant with respect to HD in Figure 3.

We have updated Figure 4 to highlight the main point that there are inter-module DC connections driven by hub genes like FAM59B. We've done this by showing all inter-module DC connections between the AD modules AD M39 and AD M26, and removed the intra-module DC connections not related to this main point. Please also note that the AD and HD modules are distinct, so the AD M39 module is distinct from the HD M39 module.

Figure 5a: It seems odd that around half of this physical interaction network would be LOC interactions and the other half would be GOC interactions, and that the physical interactions (black lines) appear to mostly be in the center, connecting the LOC and GOC parts of the graph. Would this suggest that normal interactions are likely not to be LOC or GOC in disease, but that the LOC and GOC networks are deviations from such normal interactions (or something to that effect)? This concern appears to be somewhat addressed on pages 17-19, but it is still a bit unclear, and I think this is an important point.

The non-overlap between DC and physical interactions in the aligned subnetwork is an important and surprising finding, yet compatible with our knowledge of how genes regulate each other. Basically, DC (LOC or GOC) interactions are more functional in nature, whereas direct binding (protein-protein or protein-DNA) interactions are more physical in nature; so different scenarios could lead to non-overlap (or overlap) between these two types of interactions. We hope to illustrate this with an idealized toy example described next.

Consider a transcription factor TF regulating two target genes A and B, and assume that a co-factor required for the functioning of this TF is lost in the disease condition. The TF-A and TF-B gene pairs would be the physical interactions in this example, and expression of these pairs would also be correlated in the control but not disease group due to direct regulation by the TF. More interestingly, the A-B pair could also be co-expressed in the control group due to their common regulation by the TF even if they do not exhibit direct physical interactions. So the DC relations in this example would be loss of co-regulation (LOC) of TF-A, TF-B, and A-B. We made many idealized assumptions here, such as strong correlation between protein and mRNA levels of the TF, no other regulators of A and B, and sufficient sample size to detect all “true” gene-gene correlations - these assumptions may not hold in the real data leading us to detect only a subset of all possible LOC pairs.

Related comment - page 19: "All these results suggest that the 242-gene subnetwork involving two interacting biological processes, loss of co-regulation in oligodendrocyte differentiation or myelination, and gain of co-regulation in chromatin organization could underlie multiple neurodegenerative diseases." This is a critical point that is buried at the end of a section several paragraphs long. I would suggest making it more clear early in the section that this is what you are getting at with figure 5A.

We incorporated this useful suggestion by splitting the long results section into two subsections

(Page 18) and highlighting the role of the aligned subnetwork in multiple diseases both at the end of the first subsection and beginning of the next subsection.

Page 17: "In this regard, it is notable that the propagation of DC relations between GSN and functionally related genes includes MAG (myelin-associated glycoprotein), a molecule that is synthesized in myelinated oligodendrocytes and localized at the axonal interface (Arroyo & Scherer, 2000; Trapp et al, 1989); GJB1, the gene encoding for Connexin 32, which also localizes in the myelinated fibers of the central nervous system (Scherer et al, 1995); and SOX10, a key transcriptional regulator of myelination in both central and peripheral nervous systems (Stolt et al, 2002)." Okay, but why is this notable? What is the hypothesis? Are you suggesting common dysregulation of oligodendrocytes in AD and HD involving the gene GSN, or something to that effect?

Common dysregulation of oligodendrocytes in AD and HD was indeed the main hypothesis we were trying to convey. We realized that reordering the flow of sentences in this paragraph could better convey this message, and we've changed it accordingly in Page 17.

Figure 6: This does not seem to warrant a figure that is identical to figure 5, with the exception of different colored nodes. Would it be possible to combine figure 5 and 6 in some way (maybe by coloring the text for genes changing in the Dnmt1 CKO mice?) or make Figure 6 supplemental? We've combined Figures 5 and 6 as suggested.

Middle of page 22: "...our observation that gain of co-expression (GOC, which may indicate increased activity) is a dominant feature in the DC network." If GOC may indicate increased activity, this would be useful to discuss much earlier in the paper. More generally, throughout the paper it would make the results much clearer if the biological interpretation of network concepts were discussed a bit more.

Viewing GOC pairs as an indication of increased functional activity is a hypothesis that may or may not be true. So we felt it was better placed in the Discussion. To clarify that it is an unconfirmed hypothesis, we've edited the relevant sentence in Discussion to: "Comparing molecular and macro-scale networks is not straightforward, however there are studies of both cell function and macro-scale networks in the human brain which suggest that gain of co-expression (GOC), a dominant feature in our DC networks, may indicate increased functional activity."

Reviewer #2:

The authors have done a truly thorough and thoughtful revision based upon the Reviewer's queries. I am quite impressed with the revised manuscript, particularly with the mouse model validation strategy, and am satisfied with virtually all of the responses in the letter and revisions in the text. We thank the reviewer's thoughtful and constructive suggestions, and appreciation of our diligent efforts on improving the manuscript following these suggestions.